 

## Registered report

psychology/behaviour

costly signalling theory, strategic choice model, evolution of cooperation, public goods game

**Author for correspondence:**
Martin Lang
e-mail: martinlang@mail.muni.cz

# Advertising cooperative phenotype through costly signals facilitates collective action

Martin Lang[1], Radim Chvaja[1,2], Benjamin Grant Purzycki[3], David Václavík[4,6] and Rostislav Staněk[5]

[1]LEVYNA, Masaryk University, Brno, Czech Republic
[2]PRIGO Open Research, PRIGO University, Havířov, Czech Republic
[3]Department for the Study of Religion, Aarhus University, Aarhus, Denmark
[4]Department for the Study of Religions, and [5]Department of Economics, Masaryk University, Brno, Czech Republic
[6]Department of Philosophy, Technical University of Liberec, Liberec, Czech Republic

ML, 0000-0002-2231-1059; BGP, 0000-0002-9595-7360

Around the world, people engage in practices that involve self-inflicted pain and apparently wasted resources. Researchers theorized that these practices help stabilize within-group cooperation by assorting individuals committed to collective action. While this proposition was previously studied using existing religious practices, we provide a controlled framework for an experimental investigation of various predictions derived from this theory. We recruited 372 university students in the Czech Republic who were randomly assigned into either a high-cost or low-cost condition and then chose to play a public goods game (PGG) either in a group that wastes money to signal commitment to high contributions in the game or to play in the group without such signals. We predicted that cooperators would assort in the high-cost revealed group and that, despite these costs, they would contribute more to the common pool and earn larger individual rewards over five iterations of PGG compared with the concealed group and participants in the low-cost condition. The results showed that the assortment of cooperators was more effective in the high-cost condition and translated into larger contributions of the remaining endowment to the common pool, but participants in the low-cost revealed group earned the most. We conclude that costly signals can serve as an imperfect assorting mechanism, but the size of the costs needs to be carefully balanced with potential benefits to be profitable.

# 1. Introduction

Finding reliable cooperative partners willing to commit to joint action is a crucial building block of human societies. While many societies ease cooperative problems by instituting norms that guide collective action and establish punitive mechanisms for norm transgression, people still vary in their willingness to obey these norms, especially when the short-term benefits of free-riding are temptingly high [1]. Given that willingness to cooperate in joint tasks is a hidden trait, it cannot be observed directly and might be faked through relatively cheap verbal proclamations. The potential for deception, therefore, presents a problem for the assortment of cooperative partners. Deceitful proclamations may effectively break down collective action if uncommitted individuals verbally fake their commitment and free-ride on the collective efforts. How can committed cooperators reliably recognize each other?

One answer to this conundrum is offered by the strategic choice model [2–4] designed to explain non-human animals' exaggerated phenotypes such as stotting in Thomson's gazelles [5] or the elongated upper tail coverts of the peacock's train [6]. The model asserts that the exaggeration of certain traits reliably signals hidden genetic quality that is otherwise unobservable. Based on the conflict of interest between the sender (who wishes to exaggerate the signal) and the receiver (who wishes to reliably assess the quality of the sender; [7]), the strategic choice model predicts that high-quality signallers benefit from honestly signalling their hidden quality through phenotypes that are exaggerated at a specific cost ('handicap' or 'strategic cost'; [8]), which is not affordable to low-quality signallers. For example, while the elongated tail feathers of male barn swallows handicap their flying ability, they were shown to correlate with underlying genetic quality, increased mating opportunities and reproductive success [9]. Whereas exaggerated phenotypes are not the only means to stabilize reliable communication [10,11], the mathematical formalization of the strategic choice model suggests that the differential fitness pay-offs of signalling through making phenotypes unnecessarily costly is evolutionarily stable and separates signallers based on the quality of their hidden phenotype [4,12]. Using these models, researchers identified added strategic costs in various human behavioural patterns such as meat sharing [13,14], blood donations [15], or subsistence activity, which were assessed using costly signalling theory (CST; [16–18]). Under the auspices of CST, behaviours that ostensibly decrease individual fitness in the short term (e.g. meat sharing) reliably signal hidden qualities (e.g. hunting skills), and advertisement of these qualities provides long-term fitness benefits in the form of increased mating and cooperative opportunities [16,19].

Interestingly, the application of CST to the problem of communicating a commitment to cooperative collective action—'a hidden cooperative phenotype' [1]—has been almost exclusively restricted to the context of ritual behaviour [20–22]. Indeed, it has been long recognized that ritual practices such as penile subincision of Aboriginal Australians [23] or Chukchi sacrifice of herd animals [24], together with the cross-cultural omnipresence of regular ritual gatherings where people spend time and energy, test the reliability of group members [25]. The CST of ritual (henceforth CSTR) suggests that ritual practices may be understood as signals of commitment to cooperative norms that guide collective action. The intensity of the signal may range from extreme signals such as self-mutilation to subtle signals such as attending weekly ritual gatherings [26,27]. According to CSTR, these signals, alongside other evolved mechanisms such as supernatural punishment [28–30], help mitigate problems of cooperation, such as whom to trust, accountability and collective-action maintenance. Ritual practices serve as a communication platform that offers individuals committed to collective action to truthfully express their hidden cooperative phenotype (often through expressing commitment to a supernatural deity or similar group symbols representing the group's cooperative norms), effectively separating truly committed individuals from potential free-riders [20]. Collective rituals and similar religious practices provide both a public arena and a shared code for communication of hidden cooperative phenotype [30]. By binding specific material and energetic costs to ritual performance, the hidden quality of commitment to cooperative norms materializes into physical signals that the receivers may rely upon [31,32].

Several converging lines of empirical evidence support the basic premises of the CSTR model and indicate that (i) sending costly signals (that is, performing costly rituals) correlates with hidden qualities (i.e. a cooperative phenotype) and that (ii) signal receivers understand the signal and act upon the received information. First, Sosis & Ruffle [33,34] found that self-reported participation in a public ritual of religious kibbutz members in Israel predicted contributions to the common pool in the public goods game (PGG), and Soler [35] observed a similar relationship in her work with the members of Brazilian Candomblé groups. By analysing social support networks of two villages in

South India, Power [36] showed that participants who regularly worship in a local temple and carry out public religious acts were more likely to be asked for help by other community members and more likely to provide help. Moreover, Xygalatas *et al.* [37] found that charitable contributions to the local temple after performing an extreme ritual of Thaipoosam Kavadi in Mauritius were predicted by the intensity of participation as well as by self-perceived pain suffered during the ritual. Furthermore, in another study in South India, Power [38] found that regular worship increased the chances of being nominated as generous and devout and that performing costly religious activities was associated with nominations for devoutness and for giving good advice. In her study of Brazilian Candomblé, Soler [35] also found that self-reported intensity of costly signalling predicted the reported number of cooperative offers. Finally, using fictional characters varying on the performance of costly acts showed that costly religious signallers are trusted more [39], even across religious traditions [40,41].

However, while these studies provide valuable support for CSTR by harnessing existing religious practices in various cultures, they usually cannot separate costly signals of commitment from other tangled factors and motivations underlying participation in these rituals such as personal vows to superhuman agents, anxiety management [42–44], or health improvement [27,45]. Nor can these studies disentangle the complex causal chains of religious systems that may affect cooperation [46,47]. Thus, it is not clear whether participation in religious practices is the primary driver of cooperative behaviour or whether it is the signalled cooperative phenotype driving the cooperative outputs. We aim to study the latter. In comparison with field studies, two laboratory studies using PGG suggest that simulated charity contribution and voluntary tax-paying may serve as a signal of prosocial intentions [48,49], providing preliminary evidence for the existence of cooperative-intention signalling. Nevertheless, these studies did not investigate how voluntarily undergoing self-harming (in terms of resources) rather than prosocial acts may serve this function. Furthermore, there are important caveats when applying the strategic choice model on cooperative signalling that previous studies failed to consider.

Since performing a painful ritual or sacrificing livestock is not directly and immutably linked to underlying genetic quality, the lack of commitment to collective action does not preclude ritual performance. Even free-riders may sacrifice their resources if the benefit of subsequent interaction with other ritual practitioners would offset these costs. To overcome this impasse, we suggest two possible solutions. First, based on the model introduced by Roberts [50], we propose that costly signalling of a cooperative phenotype would be stable only in iterative cooperative interactions such that the cost could not be recovered after the first interaction. If a free-rider defects during the first cooperative interaction, they would not compensate the cost of the signal but would be prohibited from other interactions (or collective action of the whole group would fail). Costly signalling may, therefore, be understood as signalling long-term cooperative intentions [50]. However, while Roberts [50] models costly signalling on the example of unspecific helping, we argue that ritual behaviour provides signals directed at cooperative norms, which crucially regulate collective action (as opposed to simple helping). The second model accounting for the discrepancy between the non-human animal and human signalling was proposed by Sosis [20], who argues that committed and uncommitted individuals differ in the *perception of cost*s associated with ritual practices. While committed members discount the costs such that the cost/benefit ratio of participation in ritual activities appears positive, the reverse is true for free-riders who differentially weigh alternative behavioural choices [51] (e.g. individually pursuing monetary gain in an unconstrained group). The differential perception of costs arises through socialization processes that, in interaction with genetically inherited traits, give rise to the cooperative phenotype. According to Sosis' model [20], the differential perception of costs would stabilize costly signals even for one-shot interactions (i.e. free-riders would perceive those as too costly).

In the current study, we test the relationship between the presence of a cooperative phenotype, willingness to send costly signals to assort with other cooperators, and the resulting level of within-group cooperation. Hailing from the strategic choice model [4] and Roberts' [50] and Sosis' [20] extensions of this model, we test four primary hypotheses. First, we test whether participants high on the cooperative phenotype would elect to reveal their hidden cooperative phenotype by sacrificing a substantial part of their resources to reliably signal their commitment to the group (H1). Second, we assess whether participants low on the cooperative phenotype would perceive this signal as too costly and refuse to send the signal (H2). Third, we investigate whether revealing the hidden cooperative phenotype positively correlates with the quality of the phenotype. That is, we examine whether participants who chose to send the costly signal would adhere to cooperative norms and contribute more to the common pool in PGG compared with participants who did not send the signal (H3). Finally, we assess whether the heightened adherence to cooperative norms and mutual assurance

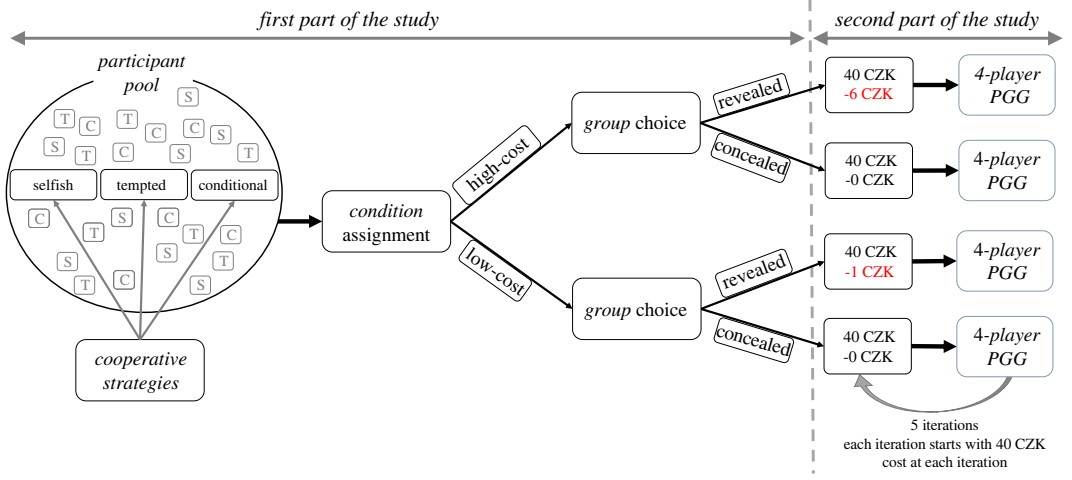

**Figure 1.** Overview of the design. In the first part of the study, participants filled out a survey on demographic variables, trained on PGG and selected their conditional choices in the pre-experiment PGG to assess their *cooperative strategy*. Next, they were randomly assigned to either the *low-* or *high-cost conditions* and subsequently selected whether they wanted to play PGG in the *revealed* or *concealed* group. In the second part of the study (approx. a week later), four participants in a given group were endowed with 40 CZK and played five PGG iterations.

through costly ritual signals would stabilize group cooperation such that individuals in groups with costly signals would earn larger monetary rewards compared with groups without signals (H4). (See §2.5 for two additional assumption checks regarding the presence of a conflict of interest and forcing uncommitted individualists to pay the signalling cost.)

To test these hypotheses, we designed a between-subjects study where we first obtained behavioural data on the expression of the cooperative phenotype using an economic game designed by Fischbacher, Gächter and Fehr (henceforth FGF) [52]. This procedure allowed us to categorize participants into three types related to their cooperative strategy (selfish, tempted and cooperative). Next, participants were randomly assigned to either a high-cost or low-cost condition. In both conditions, participants chose whether to sacrifice part of their monetary endowment to play an iterated PGG in a group with other costly signallers (the 'revealed' group) or whether to keep the total endowment and play an iterated PGG with other players who decided to 'conceal' their phenotype. The description of both groups stipulated that members of the group are expected to contribute as much as possible to the common pool (a group norm). Choosing the revealed group in the high-cost condition was associated with sacrificing 15% of the monetary endowment before each PGG iteration to signal a commitment to this norm, while only 2.5% of the endowment was sacrificed in the low-cost condition (figure 1 for a graphical overview of the design). We expected that our four main hypotheses would be supported only in the high-cost scenario, pointing to the causal role of *costly* signals.

Before data collection, we surmised that if our results would not support H1–H4 (i.e. no difference between the revealed and concealed groups in the high-cost condition), it would suggest that the effects of costly signals may be observed only in the context of group competition, which heightens the need for within-group cooperation [53,54]. Alternatively, detecting a difference between the revealed and concealed groups in both the high- and low-cost conditions would mean that even low costs are sufficient to stabilize cooperation and that the CST needs to further investigate human-specific psychology [55]. We further assumed that if H1 would not be supported, but H3–H4 would be supported, then we would conclude that the assortment of cooperators does not depend on the cooperative phenotype, and costly action is a method to induce cooperation in others by making one's choices visible [56]. Likewise, not supporting H2 but supporting H3–H4 would indicate that the cooperative phenotype does not affect the perception of costs (per Sosis' model [20]). Conversely, if H1–H2 would be supported, but H3–H4 would not be supported, we presumed that the cooperative phenotype might successfully separate signallers from non-signallers, but signals are not strong enough to stabilize cooperation above the baseline levels. For more details, see table 1.

A pilot study testing the feasibility of this approach with hypothetical PGG scenarios and high costs revealed that higher scores on a cooperative phenotype scale positively predicted the probability of sending the costly signal (H1). In comparison, lower scores on this scale predicted the probability of

**Table 1.** Overview of planned analysis and interpretation.

| | | |
|---|---|---|
| **(H1)** | *hypothesis* | The positive difference between selfish individuals and cooperators in the probability of choosing the revealed group would be larger in the high-cost relative to the low-cost condition. We remain agnostic about comparisons of selfish individuals with tempted individuals. |
| | *analytical model* | $y_i \sim \text{Binomial}(n_i,\, p_i)$ <br><br> $\text{logit}(p_i) = \alpha + \beta_{1[LC\ versus\ HC]} + \beta_{2[CC\ versus\ TC]} + \beta_{3[CC\ versus\ SF]} + \beta_{4[HC\,*\,TC]} + \beta_{5[HC\,*\,SF]}$ <br><br> $y_i$ is individuals' ($i$) choice of the revealed group ($n_i = 1$ since there is only one choice). $p_i$ is the probability of choosing the revealed group. $\alpha$ is a fixed intercept, and $\beta_1$ is the parameter for the effect of comparison between the low-cost and high-cost conditions for conditional cooperators. $\beta_2$ and $\beta_3$ are the parameters for the effect of comparison between the conditional cooperators and tempted individuals and between the conditional cooperators and individuals with selfish strategy, respectively, in the low-cost condition. $\beta_4$ and $\beta_5$ are the interaction terms comparing the effects of the three cooperative strategies between conditions. <br><br> R model: <br> $\text{glm}(group \sim coop\_type * condition, family\ =\ 'binomial')$ |
| | *interpretation* | Pilot data suggest a semi-separating equilibrium whereby some cooperators may choose to hide their phenotype. If no substantial effect would be detected, we would investigate whether the separation process was convoluted by cooperators choosing the concealed group or by selfish individuals choosing the revealed group. In combination with support for H3–H4, we could conclude that the assortment of cooperators does not depend on the cooperative phenotype, and costly action is a method to induce cooperation in others by making one's choices visible. |
| **(H2)** | *hypothesis* | The negative difference between individuals playing selfish and cooperative strategies in the probability of stating that the costs in the revealed group were too high/unreasonable would be larger in the high-cost relative to the low-cost condition. We remain agnostic about comparisons of selfish strategies with tempted strategies. |
| | *analytical model* | $y_i \sim \text{Binomial}(n_i, p_i)$ <br><br> $\text{logit}(p_i) = \alpha + \beta_{1[LC\ versus\ HC]} + \beta_{2[CC\ versus\ TC]} + \beta_{3[CC\ versus\ SF]} + \beta_{4[HC\,*\,TC]} + \beta_{5[HC\,*\,SF]}$ <br><br> $y_i$ is whether ($= 1$) or not ($= 0$) individuals ($i$) mention that costs are too high/unreasonable in the revealed group ($n_i = 1$ since there is only one choice). $p_i$ is the probability of mentioning high costs. $\alpha$ is a fixed intercept, and $\beta_1$ is the parameter for the effect of comparison between the low-cost and high-cost conditions for conditional cooperators. $\beta_2$ and $\beta_3$ are the parameters for the effect of comparison between the conditional cooperators and tempted individuals and between the conditional cooperators and individuals with selfish strategy, respectively, in the low-cost condition. $\beta_4$ and $\beta_5$ are the interaction terms comparing the effects of the three cooperative strategies between conditions. <br><br> R model: <br> glm(high_cost $\sim$ contribution $\times$ condition, family = 'binomial') |
| | *interpretation* | If cooperators would separate based on their hidden phenotype, but individuals playing selfish strategy would not be deterred by cost in the high-cost condition, we would interpret this finding as possibly unconscious motivation for joining either of the two groups and re-analyse the free-list data regarding the reasons for choosing the group. In combination with support for H3–H4, we would conclude that the cooperative phenotype does not affect the perception of costs. |

| (H3) | hypothesis | The positive difference between participants in the concealed group and participants in the revealed group in the portion of their endowment contributed to the common pool would be smaller in the low-cost compared with the high-cost condition. |
|---|---|---|
| | analytical model | $y_{ri} \sim \beta(\mu_i, \phi)$ <br> $\text{logit}(\mu_i) = \alpha + \omega_{[m(i)]} + \beta_{1[LC \text{ versus } LC]} + \beta_{2[CN \text{ versus } RV]} + \beta_{3[HC*RV]}$ <br><br> $y_{ri}$ is the proportion of endowment contributed in PGG iteration $r$ by an individual $i$. $\alpha$ is a fixed intercept, $\omega_{[m(i)]}$ is a varying intercept for individual participants across multiple measures ($m$). $\beta_1$ is the parameter for the effect of comparison between the low-cost and high-cost conditions for the concealed group. $\beta_2$ is the parameter for the effect of comparison between the concealed and revealed groups in the low-cost condition. $\beta_3$ is the interaction term between condition and group. The parameters of the assumed beta distribution comprise $p_i$ representing location (i.e. proportion of send to kept endowment), and $\phi$ denotes dispersion. <br><br> R model: <br> glmmTMB(proportion_contributed $\sim$ group $\times$ condition + (1\|ID), family = 'beta_family') If the data would contain >1/3 of zeros and ones, we would also fit a zero-or-one inflated beta (ZOIB) model. |
| | interpretation | If the 95% CIs for the interaction term would include zero and the assumed separation process functional, we would investigate in a follow-up study whether this is due to the signal being non-functional (no difference between the revealed and concealed groups) or the functional assortment of cooperators in the low-cost condition. If the former would be true, we would continue our investigation by designing an experimental procedure where two groups would compete against each other (following the suggestion that costly signalling intensifies during between-group conflict). |
| (H4) | hypothesis | Participants in the revealed group would earn more than participants in the concealed group in the high-cost condition. This difference would not apply to the low-cost condition. |
| | analytical model | $y_i \sim \text{Poisson}(\lambda_i)$ <br> $\log(\lambda_i) = \alpha + \beta_{1[LC \text{ versus } HC]} + \beta_{2[CN \text{ versus } RV]} + \beta_{3[HC*RV]}$ <br><br> $\lambda_i$ is the expected earned amount after five iterations by an individual $i$. $\alpha$ is a fixed intercept. $\beta_1$ is the parameter for the effect of comparison between the low-cost and high-cost conditions for the concealed group. $\beta_2$ is the parameter for the effect of comparison between the concealed and revealed groups in the low-cost condition. $\beta_3$ is the interaction term between condition and group. <br><br> R model: <br> glmmTMB (earned $\sim$ group $\times$ condition, family = 'poisson') If the Poisson model would reveal overdispersion, we would fit a negative binomial model instead. If the residuals would be distributed normally, we would also fit an OLS model. |
| | interpretation | If the 95% CIs for the interaction term would include zero and the assumed separation process functional, we would investigate the trends in earnings over individual iterations to estimate how many iterations would be needed to support H4. If no trends would be detected, we would proceed in the same steps as in the case of not supported H3. |

mentioning that the cost of the signal is too high (H2). Notably, participants who chose to send the costly signal reported that they would send a higher portion of their remaining endowment to the common pool in a hypothetical one-shot PGG (H3). However, comparing the hypothetical earnings between the revealed and concealed groups indicated no substantial difference in their earnings (H4 not supported). See §2.5. While generally supportive of the proposed framework, these results need to be

bolstered by an actual behavioural study on a high-powered sample, by an iterated PGG that better reflects the dynamics of real-world collective action where cooperative interactions between specific individuals are often repeated [57] and by manipulation of the costliness of the signal.

# 2. Methods

## 2.1. Ethics information

This study was approved by the Ethics Committee for Research at Masaryk University. Participants provided informed consent and received at minimum 50 CZK (1.7 GBP) as a show-up fee plus any amount they earned in PGG.

## 2.2. Design

The study used a between-subjects and double-blinded design. Research assistants and participants were blind to our hypotheses. However, note that we did not use deception, and participants knew that they self-selected into one of the two groups. We offered participation in this study to students from the subject pool at Masaryk University, Czech Republic.

### 2.2.1. First part of the study (individual)

If interested, participants were first asked to fill out an online survey including basic demographic information (age, sex, economic status; note that we did not plan to use these variables in our pre-registered analyses due to the homogeneity of our population, but they may be used for further exploration). Next, participants read instructions on playing a one-shot PGG with the following parameters: an initial endowment of 100 CZK, anonymously invested in an interval of 5 CZK simultaneously with the other three participants). The sum in the common pool was doubled, and the earnings equally distributed among the players. Participants were provided with three examples of how the game could evolve to affect players' earnings. We asked participants to fill out a fourth hypothetical scenario to test their understanding of PGG rules. If they failed to pass the understanding check, the rules of the game were explained again, and participants were offered to answer once more. If they failed the second understanding check, they were not invited to participate in the second portion of the study (but paid the show-up fee).

After passing the understanding check, participants were asked to make unconditional and conditional decisions in a one-shot PGG per the FGF [52] procedure. Specifically, participants made one decision on contributing to PGG out of 100 CZK endowment (in an interval of 5 CZK) without knowing how much the other three players contributed (unconditional). Then, participants made 21 conditional decisions based on the average contribution of the other three players (rounded to the nearest multiple of five; i.e. for an average contribution of 0, 5, 10 … 100). After the end of the first part of the study, participants were randomly paired with three other players, and one participant from each group was randomly selected as the relevant person for the conditional decision. PGG pay-offs were calculated based on the unconditional contributions of the remaining three players and the respective conditional contribution of the selected player. As such, participants were motivated to make genuine decisions because these decisions impacted their earnings. The earnings were known to participants and paid only after the second part of the study. For more details, see FGF [52]. Using the conditional choices, we categorized participants into three different cooperative strategies that we assumed reflect their hidden cooperative phenotype (see below). For the complete survey, see materials at the Open Science Framework (OSF) repository.

After making their choices in the FGF procedure, participants were introduced to the second part of the study conducted approximately one week later. Specifically, participants were told that they will be endowed with 40 CZK and play PGG simultaneously with three other individuals. Participants were informed that they will decide how much of their endowment to send into the common pool while not knowing how much the others will send. After making individual contributions, money in the common pool will be multiplied by two and evenly distributed among the four group members, irrespective of their contributions. Next, they were told that they will again receive 40 CZK and play the second round of PGG with the same group members and likewise in the third, fourth and fifth round of PGG.

Crucially, participants were presented with a choice of two groups for the actual PGG play in the second portion of the study (participants played all five iterations in the same group of their choice). We call these groups a 'revealed' group and a 'concealed' group; however, participants decided between Groups X and Y.

The instructions for the X group (the 'concealed' group) were as follows:

In Group X, money invested into the common pool is multiplied by two and equally distributed among group members. It is expected that every member should contribute as much as possible to the common pool to increase the welfare of all group members; however, all contributions are anonymous. Choosing this group is not associated with a cost, and it is not necessary to demonstrate intentions regarding the size of the contribution to the common pool.

The instructions for the Y group in the high-cost condition (the 'revealed' group) were as follows:

In Group Y, money invested into the common pool is multiplied by two and equally distributed among group members. It is expected that every member should contribute as much as possible to the common pool to increase the welfare of all group members; however, all contributions are anonymous. Choosing this group is associated with a monetary cost that no one will benefit from. Specifically, members of this group will sacrifice 15% [or 2.5% in the low-cost condition] of their endowment before each PGG round (6 CZK [1 CZK in low-cost]) to demonstrate their intentions regarding the size of the contribution to the common pool.

The order of the group presentation (X or Y first) was randomized. Upon choosing either the Y or X group, participants were prompted to provide a short rationale for choosing their group and invited to participate in the second part of the study one week later. Participants were invited to the second part of the study in groups of four to interact with members of the same group (revealed or concealed). We invited participants in such a way as to approximately balance the number of women and men for each condition.

## 2.2.2. Second part of the study (group interaction)

For the second part, we invited at least one extra participant for each session to ensure that each session had four participants. If not needed, these extra participants were paid a show-up fee of 100 CZK plus any earning from the FGF played online before the experiment. If we had fewer than four participants in one session, we did not run the session. Upon joining the online testing session, participants were welcomed by a research assistant through a virtual chat platform, individually read an informed consent and reminded of the rules of PGG specific for their group.

Next, participants were introduced to a software (oTree) programmed to facilitate the multi-player iterated PGG. On a computer screen, they saw their actual earnings, and for each PGG iteration, they inputted the percentage of their endowment they wanted to contribute to the common pool. After each PGG iteration, the software summed all players' contributions, multiplied them by two, and equally distributed this product between the group members, updating their earnings. Specifically, in the high-cost condition, participants in both groups started each PGG round with an initial endowment of 40 CZK (approx. 1.3 GBP), and participants in the revealed group immediately lost 15% of their endowment, i.e. 6 CZK. Using their remaining endowment, participants in both groups made first iteration PGG decisions and learned about others players' contributions and their current earnings. Before the second PGG round, each participant in the revealed group again lost 6 CZK from their 40 CZK endowment for the second round, and the same procedure was repeated for the third, fourth and fifth PGG iteration (after each iteration the participant would see other players' anonymous contributions). Upon finishing the gameplay, participants were paid out their earned sum after five iterations plus a show-up fee and earnings from FGF. The maximum earning for full cooperation in the revealed group was set at 408 CZK, while for the concealed group at 480 CZK in the high-cost condition. The maximum earning for playing the selfish strategy while the other three players would unconditionally cooperate was set at 425 CZK in the revealed group and 500 CZK in the concealed group.

In the low-cost condition, this procedure was identical except that participants chose between a concealed group without any signal and a revealed group that would sacrifice 2.5% of their endowment, that is, 1 CZK. The maximum earning for full cooperation in the revealed group was set at 468 CZK, while for the concealed group at 480 CZK. The maximum earning for playing the selfish strategy while the other three players would unconditionally cooperate was set at 488 CZK in the revealed group and 500 CZK in the concealed group (see OSF materials for these calculations).

## 2.3. Sampling

Participants were recruited from a student participant pool at Masaryk University, Czech Republic. Expectedly, sampling from this pool in our previous studies [46,47] revealed a young (mean age = 24) and secular sample where the modal answer on religiosity was 'not religious', and the modal answer on ritual participation was 'never/not often'. Hence, testing our hypotheses on a population largely unengaged with costly religious signals should present a strong test of our hypothesis.

For both conditions, we invited participants separately for the revealed- and concealed-group sessions to balance the ratio between the revealed and concealed groups and the ratio of women and men in each group. We planned to recruit 160 participants for each condition: approximately 80 per group and four per session (for a grand total of 320 participants). Participants who filled out the first portion of the study but did not take part in the second part of the study were paid a show-up fee and FGF earnings. The planned sample size was based on the cost/benefit ratio of power analyses for our four hypotheses.

To assess the expected power of the main planned statistical tests for H1–H4, we specified three estimated effect sizes for the interaction between cooperative strategy and condition (H1–H2) and for the interaction between chosen group and condition (H3–H4). Specifically, for each hypothesis, we expected no effects of the main predictors (strategy/group) in the low-cost condition and varied the effect sizes for the high-cost condition based on pilot data and theoretical expectations (see electronic supplementary material, figure S1 for expected effects). Next, we used the command *powerSim* from the *simr* package [58] in R to simulate the planned statistical models for various sample sizes. *simr* uses Monte Carlo simulation with pre-specified effect size and variance explained by varying intercepts (and other relevant parameters for other distributions) to re-fit the planned statistical model a specified number of times, assessing the binomial ratio of models with significant/non-significant results (at significance level $\alpha$ = 0.05). We used 1000 Monte Carlo simulations to simulate the expected differences in slopes between conditions for each hypothesis for sample sizes ranging from 40 per condition to 240 per condition (in the steps of 20). The results of these simulations (with 95% CIs) are plotted in electronic supplementary material, figure S1, suggesting that 160 participants per condition should allow us to detect moderate effect sizes of the specific interactions with greater than 80% power for all four hypotheses.

Since most of the questionnaire data were collected online, we did not expect missing data. Since our primary outcome and predictor variables are bounded, we did not expect to detect any outliers, and we used appropriate statistical techniques to account for participants scoring on the boundaries of possible data distribution (see Analysis). Finally, we planned to exclude participants who did not pass an understanding check (specified above) but this was not the case for any participant in the second round. Likewise, we planned to screen participants' reasons for choosing either of the groups and exclude those whose responses indicating a misunderstanding of the group definitions. We did not exclude any participant on this basis. There were no additional exclusion criteria.

## 2.4. Analysis

Analyses were conducted in R [59] (R version for the presented analyses: 3.6.3). First, we categorized participants into three cooperative strategies based on their play of the FGF version of PGG. Namely, participants playing a cooperative strategy (corresponding to the cooperative phenotype), tempted individuals (cooperate if the temptation to free-ride is low but free-ride if benefits are high), and individuals playing a selfish strategy (always free-ride). To this end, we fitted a finite mixture model to our FGF data using the function *flexmix* from the flexmix package [60]. This function estimates distributional parameters for each of the three cooperative strategies and then classifies participants into one of those strategies (for an example, see Chen & Fischbacher [61]). We also coded participants' responses to the open question on the reasons for choosing their particular group, searching for words such as 'waste', 'loss' or 'unnecessary' concerning the signal cost, which would indicate that the signal was perceived as too costly (see electronic supplementary material, section S2.2 for examples from the pilot data). Two independent coders blind to our hypotheses coded participants' answers with 87% agreement. The first author of this study arbitrated the 13% of responses on which the two coders did not agree.

To analyse the mean contribution to the common pool in each PGG, we first calculated the percentage contributed to the common pool from the remaining endowment, accounting for various costs between groups and conditions and the fact that there was a limit on the minimum and maximum contribution.

The overall earning in PGG for each participant was calculated as a sum of individual earnings in every PGG iteration.

Next, we tested our hypotheses using a generalized linear mixed model (GLMM) framework, accounting for the specific data-generation process and hierarchical structure tailored to each hypothesis. The first hypothesis was assessed using logistic regression with the probability of choosing the revealed group as the outcome variable, and cooperative strategy (selfish versus tempted versus cooperators) interacted with condition (low-cost versus high-cost) as the predictor variables. The second hypothesis was assessed using a binomial regression where the outcome variable was the probability of mentioning that costs were too high in the revealed group. The predictors comprised cooperative strategies (selfish versus tempted versus cooperators) interacted with condition (high-cost versus low-cost). The third hypothesis was planned to be assessed using a beta regression to account for the typical structure of percentage data [62,63], where the proportion of the endowment contributed to the common pool across the five PGG iterations would comprise the outcome variable and group (concealed versus revealed) interacted with condition (low-cost versus high-cost) the main predictor variable. However, we also planned that if more than 1/3 of the PGG contributions would contain 0 or 1, we would fit a zero-or-one-inflated beta model. Since this was the case (see Results), we fitted the zero-or-one inflated beta (ZOIB) model using the *gamlss* package [64]. We adjusted the model estimates for the fact that individuals were nested within the five PGG iterations. Finally, H4 was assessed by a structurally similar model as H3; only the dependent variable was the sum individual earnings in CZK after all five PGG iterations. We planned to use Poisson regression to account for the fact that our data were bounded by minimum and maximum earnings. However, we also planned that if the Poisson model would display overdispersion (as suggested by pilot data), we would opt for a negative binomial model instead and that if the data would be approximately normally distributed, we would consider using an ordinary least-square regression (OLS) for a more straightforward result presentation. As we detected overdispersion, we fitted a negative binomial model. A detailed overview of the statistical tests assessing each hypothesis can be found in table 1 and the electronic supplementary material, R code.

## 2.5. Pilot data

To assess the feasibility of the planned procedure, we conducted two online pilot studies (henceforth Pilot 1 and Pilot 2) with the Czech student population. Participants for the pilot studies were recruited through advertisement at various student groups on Facebook and asked for help testing a new study. No compensation was offered for participation. For Pilot 1, we recruited 89 participants (63 women; $M_{age} = 23.9$) and for Pilot 2, we recruited 91 participants (68 women; $M_{age} = 24.5$).

### 2.5.1. Pilot design

Since this is was an online study, we assessed the cooperative phenotype using a cooperative values scale adapted from Peysakhovich *et al*. [1] rather than the cooperative strategy planned for the actual experiment (see electronic supplementary material, section S2.1 for the specific items and reliability analysis). Note that we did not plan to use this scale as a predictor in the actual experiment. Next, we explained the rules of PGG and tested participants' understanding of the PGG rules (see §2.2). Participants who failed the second understanding check were excluded from the analysis (three participants in Pilot 1 and five participants in Pilot 2). We also excluded participants who did not finish the survey (three participants in Pilot 2), and one participant who reported being 96 years old.

After explaining the rules of PGG, participants were asked to imagine three hypothetical PGG scenarios played with three other players:

#### 2.5.1.1. First scenario

In the first scenario, participants were asked to imagine receiving an endowment of 200 CZK and playing one-shot PGG as the last player, that is, after knowing how much other hypothetical players contributed to the common pool. This scenario aimed to test an assumption of the signalling theory that people vary in cooperative affordances (conditional cooperators versus selfish individuals). That is, we tested Assumption Check 1 (AC1), stating that selfish individuals often defect collective action for personal benefits while conditional cooperators mainly contribute to collective action for their mutual benefit. We varied the contributions of other hypothetical players such that the remaining three players

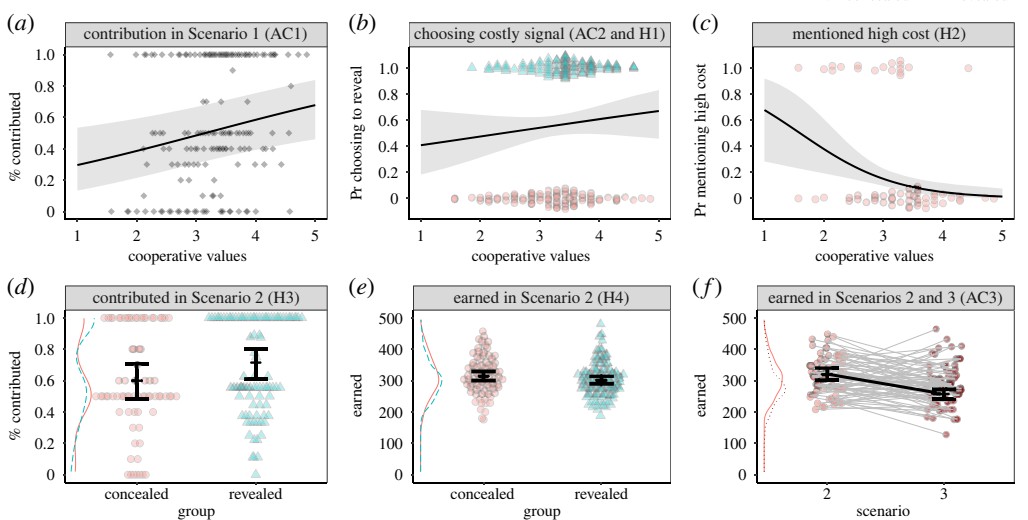

**Figure 2.** Overview of pilot data. Reporting higher cooperative values increased reported contributions to the common pool in PGG Scenario 1 (*a*) as well as increased the probability of choosing the revealed group (*b*). Lower reported cooperative values positively predicted mentioning costly entrance fee in the revealed group as too high (*c*). Participants in the revealed group reported that they would send a higher percentage of their endowment to the common pool in PGG Scenario 2 (*d*), but this would not lead to higher earnings (*e*). Finally, forcing participants in the concealed group to make the costly signal would decrease their earnings (*f*). Black lines are regression estimates with 95% CIs. Figures *d–f* also contain density plots for the respective comparisons.

contributed either their entire endowment (i.e. 200 CZK), part of their endowment (80 CZK), or variable sums (200, 0 and 20 CZK).

### 2.5.1.2. Second scenario

In the second scenario, participants were presented with two hypothetical groups they could join for yet another PGG in which they would again receive 200 CZK endowment and contribute simultaneously with the other three players to the common pool. These hypothetical groups afforded either to reveal participants' commitment to contribute to the common pool at a cost (20% of their endowment in Pilot 1 and 10% of their endowment in Pilot 2) or to save the money and play in a group that conceals intentions (see §2.2 for definitions of the two groups). Given the assumed variation of the cooperative phenotype in the population [1,65], we expected that participants would self-select roughly 50/50 in the concealed and revealed groups (Assumption Check 2; AC2). After choosing either of the two groups, we asked participants how much they would contribute to the common pool and to give a reason for choosing the specific group, testing H1–H4.

### 2.5.1.3. Third scenario

Finally, to test another assumption of the signalling model in these hypothetical scenarios, we included a third scenario in Pilot 1 where we asked participants who played in the revealed group to imagine playing in the concealed group and vice versa. The purpose of this manipulation was to examine whether the costly signal would indeed be too costly for the uncommitted individualists, hypothetically present in the concealed group. Thus, the third Assumption Check (AC3) stated that participants who chose the concealed group in the second PGG scenario should earn less when forced to signal norm commitment in the revealed group in the third PGG scenario. The results of the pilot test are described in verbatim below and plotted in figure 2. Note that we used GLMM in the pilot-data analysis analogously to models in table 1 (see also electronic supplementary material, R code).

### 2.5.2. Pilot results

The results of the first PGG scenario collapsed across both pilots, and three contribution schemas suggest that participants scoring higher on the cooperative values scale reported that they would contribute a larger portion of their endowment to the common pool ($\beta = 0.40$, 95% CI = [0.11, 0.69]; supporting AC1). Scoring 'one' on the cooperative values scale was associated with a reported contribution of

30% of the endowment, while scoring 'five' was predicted to yield contributions at 68% of the endowment. Furthermore, signalling strategies were roughly equally represented (*n revealed* = 94, *n concealed* = 74; supporting AC2), and higher scores on the cooperative values scales positively predicted the probability of choosing the revealed group (supporting H1), albeit this effect was not precise and 95% CIs contained zero ($\beta$ = 0.27, 95% CI = [−0.20, 0.74]). The lowest score on the cooperative values scale predicted a 41% probability of choosing the revealed group, while the maximum score was associated with 67% probability. This imperfect separation is further explored in the electronic supplementary material, section S2.3. Supporting H2, higher scores on the cooperative values scale negatively predicted the probability of mentioning that the cost of the revealed group was too high ($\beta$ = −1.24, 95% CI = [−2.04, −0.44]) when prompted to explain why they chose to play in the concealed group (estimated 68% probability for cooperative values score of 'one').

Participants in the concealed group reported that they would contribute a smaller proportion of their remaining endowment compared with participants in the revealed group (60% versus 72%), supporting H3 ($\beta$ = 0.52, 95% CI = [0.14, 0.90]). However, when assessing how much players in each group would earn after summing contributions in hypothetical sessions with other players, overall estimated earnings were higher in the concealed (315) compared with the revealed (302) group ($\beta$ = −0.04, 95% CI = [−0.10, 0.02]). While this result does not support H4, the difference between groups was not precisely estimated, and we expected that using real monetary incentives in iterated PGG would support H4 (there would be a steady decline in mean contributions in the concealed group as more members choose to free-ride in subsequent iterations, as shown by other PGG experiments [66]).

Finally, we compared the potential earnings of participants who chose the concealed group in the second scenario with their hypothetical earnings in the third scenario, where they were forced to play in the other group. Participants in the concealed group would, on average, earn less in the third scenario (321 versus 257), supporting AC3 ($\beta$ = −0.22, 95% CI = [−0.28, −0.16]). Further details on the pilot procedures and additional analyses are reported in the electronic supplementary material.

# 3. Results

## 3.1. Classification of cooperative strategies

To account for dropouts between the two stages of the study and have sufficient substitutes, we initially recruited 458 participants for the first phase of the study who were randomly assigned to the high-cost and low-cost conditions. From this pool of participants, 372 (197 women; 1 non-binary; $M_{age}$ = 23.6, s.d.$_{age}$ = 3.1) finished the first phase and were interested in taking part in the second phase of the study (we aimed for a final sample of 320 participants, 80 per each combination of group and condition). Of the 372 participants, 90 participants chose the revealed group and 99 participants chose the concealed group in the high-cost condition. In the low-cost condition, 122 participants chose the revealed group and 61 participants chose the concealed group.

We classified participants into three different cooperative types based on the strategies they played in the FGF version of PGG: cooperators, tempted cooperators and individuals playing a selfish strategy (figure 3). We assumed that these strategies should approximate the underlying cooperative phenotype. The model classified 171 participants as cooperators, 107 as tempted cooperators and 94 as playing selfishly. The ratio of prior and posterior probabilities for all three categories was greater than 0.994, suggesting a non-overlapping classification of participants [60].

## 3.2. Choosing the revealed group (H1)

We used this classification to predict the selection of concealed and revealed groups in the high- and low-cost conditions, hypothesizing that individuals playing selfishly will be less likely to choose the revealed group in the high-cost condition. The results of our binomial regression model lent support to this hypothesis, showing that compared with cooperative behaviour, selfish behaviour in the conditional PGG was associated with a lower probability of choosing the revealed group in the high-cost condition ($\beta$ = −1.05, 95% CI = [−1.78, −0.32]). As predicted, this difference was smaller in the low-cost condition, although the 95% confidence intervals of this interaction included zero ($\beta_{interaction}$ = 0.80, 95% CI = [−0.25, 1.85]). Since most of the probability mass was positive, we interpret this difference as a preliminary support for H1. Looking at the high-cost condition, cooperators had a 57% chance of choosing the revealed group while this probability dropped to 46% for tempted cooperators and only to 32% for

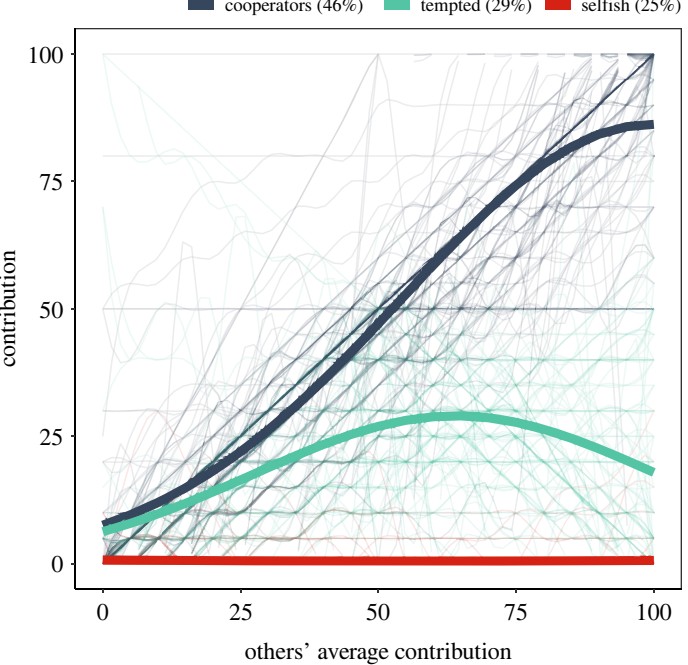

**Figure 3.** Classification of participants into three cooperative strategies. The thick lines plot the predicted values for each of the cooperative strategies, while the thin lines represent raw data colour-coded based on the specific strategies. We used a cubic spline interpolation on the raw data for easier visual reading.

**Table 2.** Beta-estimates from logistic regressions with 95% CI from testing hypothesis 1 (probability of selecting the revealed group) and hypothesis 2 (probability of mentioning high costs). The reference category is 'cooperators' for the strategy factor and 'high cost' for the condition factor. The estimates are logged odds.

|  | hypothesis 1 | hypothesis 2 |
|---|---|---|
| intercept | 0.29 | −1.89 |
| | (−0.13, 0.71) | (−2.53, −1.26) |
| strategy: tempted | −0.45 | −0.05 |
| | (−1.15, 0.24) | (−1.12, 1.01) |
| strategy: selfish | −1.05 | 1.07 |
| | (−1.78, −0.32) | (0.17, 1.96) |
| condition: low-cost | 0.42 | 0.23 |
| | (−0.21, 1.04) | (−0.65, 1.12) |
| low-cost × tempted | 0.60 | 0.08 |
| | (−0.41, 1.61) | (−1.37, 1.52) |
| low-cost × selfish | 0.80 | −1.44 |
| | (−0.25, 1.85) | (−2.87, −0.01) |
| N participants | 372 | 345 |

individuals with selfish strategy. By contrast, these probabilities were estimated at 67%, 70% and 61% in the low-cost condition. See table 2 for all estimates, figure 4a for illustration, and electronic supplementary material, R code for re-analysis of this data using raw conditional contributions for each type as predictors.

## 3.3. Differential perception of costs (H2)

We further used the classification into cooperative strategies to predict whether participants mentioned wasted resources when verbally explaining their choice of the group for the second phase. From 345

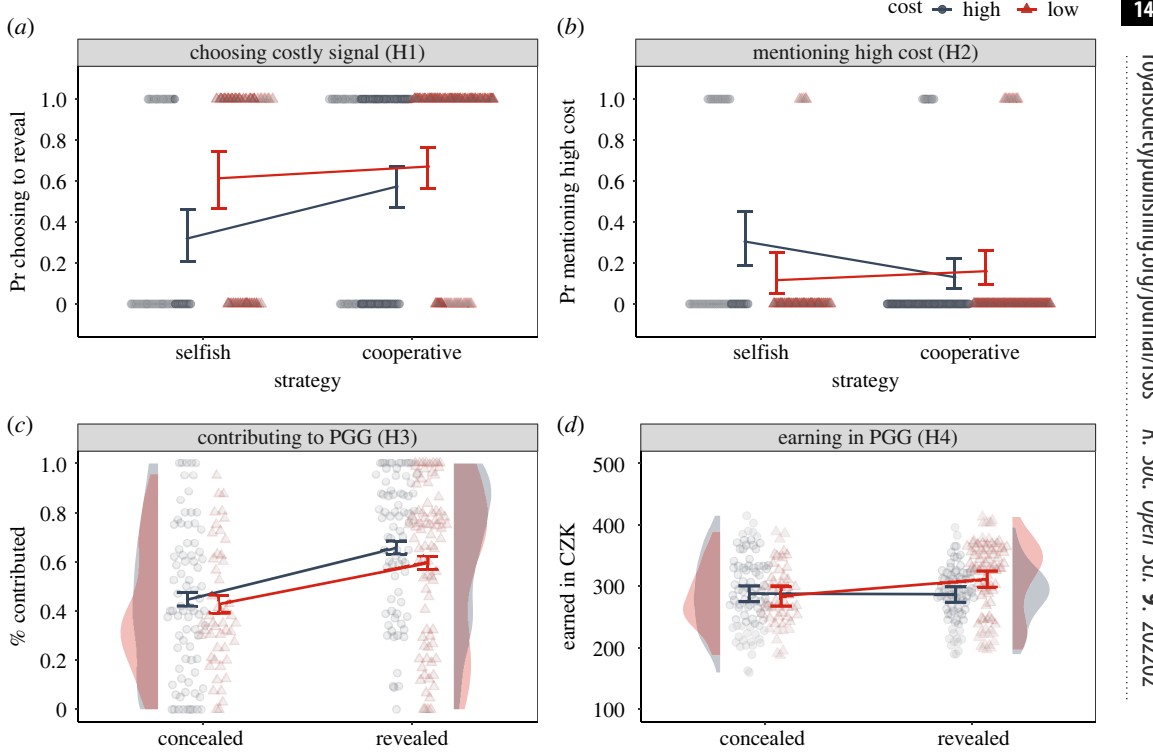

**Figure 4.** Estimated means with 95% CI plotted over raw data for the four core hypotheses. The density plots represent the distribution of raw data for each group/condition combination. Note that the estimated lines in plot) C are from a beta regression rather than ZOIB regression to include the full spectrum of the data modelled by the mixture of separate regressions in the main text (table 3). Specifically, in this graph, the 0 and 1 contributions were converted using the formula $(y' = (y(n-1) + 0.5)/n)$ where y is the transformed variable and $n$ is the sample size, such that the data could be analysed with the beta regression (this correction has negligible effects on inference). See electronic supplementary material, R code for precise estimates from this model.

participants who answered this question, 56 participants mentioned that the signal is a waste of resources. Using a binomial regression model, we found a difference in mentioning waste between individuals playing cooperative and selfish strategies in the high-cost condition ($\beta = 1.07$, 95% CI = [0.17, 1.96]) and this difference was smaller in the low-cost condition ($\beta_{\text{interaction}} = -1.44$, 95% CI = [−2.87, −0.01]). The estimated probabilities of seeing the costly signal as inefficient were 14% for cooperators, 13% for tempted and 30% for individuals playing selfishly in the high-cost condition and 16%, 16% and 12% in the low-cost condition. See table 2 for all estimates and figure 4b for illustration.

## 3.4. Contributions to the common pool (H3)

From 372 participants who proceeded to the second phase of the study, 284 actually participated (146 women; 1 non-binary; $M_{\text{age}} = 23.5$, s.d.$_{\text{age}} = 3.0$). The remaining participants were either substitutes on a given experimental session or did not show up for a session. Note that we succeeded to collect data from 80 participants in the high-cost concealed group and low-cost revealed group as planned, but we missed data from four participants in the high-cost revealed group because not enough participants showed up for an experimental session. Moreover, only 61 participants chose the concealed group in the low-cost condition, and when accounting for participants who were selected as substitutes, this group comprised 48 participants instead of 80. Nevertheless, the a priori analysis plotted in electronic supplementary material, figure S1 suggests that 284 participants should be sufficient to detect the expected effects with 80% power for H3 and 75% power for H4.

Looking at the raw contributions, participants allocated 54% of their remaining endowment on average. The average allocations were highest in the first round (62%) and lowest in the last round (36%). Figure 5 provides an illustration of raw data. Since 45% of the allocations were either 0% or 100% of the endowment, we used the zero-or-one inflated beta regression (ZOIB) that allows to infer the probability of contributing zero or one as well as the size of the mean contribution (for technical details see [67]).

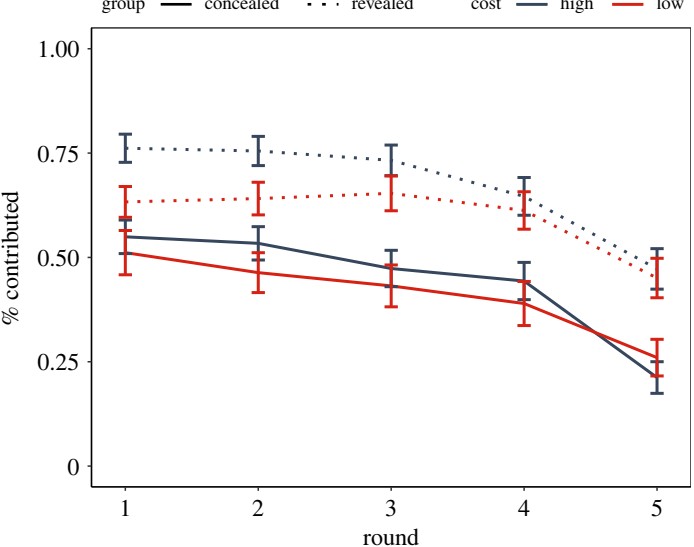

**Figure 5.** Means with SE of the proportion of remaining endowment contributed to the common pool.

**Table 3.** Beta-estimates from GLMM with 95% CI from testing hypotheses 3 (highest contributions in the high-cost revealed group) and 4 (highest earnings in the high-cost revealed group). *Note.* The reference category is 'concealed' for the group factor and 'high cost' for the condition factor. The estimates are untransformed.

|  | hypothesis 3 % sent | hypothesis 3 pr. sending 0 | hypothesis 3 pr. sending 1 | hypothesis 4 |
|---|---|---|---|---|
| intercept | −0.23 | −0.94 | −1.57 | 5.66 |
|  | (−0.33, −0.13) | (−1.23, −0.64) | (−1.92, −1.23) | (5.62, 5.67) |
| group: revealed | 0.55 | −1.44 | 1.02 | −0.003 |
|  | (0.40, 0.69) | (−1.96, −0.93) | (0.57, 1.46) | (−0.06, 0.06) |
| condition: low-cost | −0.02 | −0.63 | −0.71 | −0.01 |
|  | (−0.18, 0.13) | (−1.11, −0.15) | (−1.29, −0.13) | (−0.08, 0.06) |
| low cost × revealed | −0.21 | 0.93 | 0.31 | 0.10 |
|  | (−0.42, 0.004) | (0.19, 1.68) | (−0.39, 1.02) | (0.004, 0.19) |
| *N* participants | 284 | 284 | 284 | 284 |

For average contributions excluding zeros and ones across the five rounds of PGG, the model showed that participants in the high-cost revealed group contributed larger portions of their endowment compared with participants in the high-cost concealed group ($\beta = 0.55$, 95% CI = [0.40, 0.69]). This difference was smaller in the low-cost condition ($\beta_{\text{interaction}} = -0.21$, 95% CI = [−0.42, 0.004]). Furthermore, the parameter modelling the probability of contributing zero was smaller in the high-cost revealed group compared with the high-cost concealed group ($\beta = -1.44$, 95% CI = [−1.96, −0.93]) and this difference was again smaller in the low-cost condition ($\beta_{\text{interaction}} = 0.93$, 95% CI = [0.19, 1.68]). The estimated probabilities of contributing zero were 6% for the high-cost revealed group, 25% for the high-cost concealed group, 8% for the low-cost revealed group, and 16% for the low-cost concealed group. Finally, the parameter modelling the probability of contributing the full remaining endowment (conditioned on the probability of contributing zero) was higher in the high-cost revealed group compared with the high-cost concealed group ($\beta = 1.02$, 95% CI = [0.57, 1.46]). However, this difference was larger, albeit not reliable, in the low-cost condition ($\beta_{\text{interaction}} = 0.31$, 95% CI = [−0.39, 1.02]). While the probability of sending everything was the highest in the high-cost revealed group (34% compared with 26% in the low-cost revealed group), it was low in the low-cost concealed group (8% compared with 13% in the high-cost concealed group). Refer to table 3 for all estimates and figure 4*c* for illustration.

## 3.5. PGG earnings (H4)

For our final hypothesis test, we compared participants' earnings across conditions. Participants, on average, earned 294 CZK in PGG. Using a negative binomial regression to model count data (Poisson distribution was disqualified due to overdispersion), we found that contra our predictions there was no difference in earnings between the high-cost revealed group and the high-cost concealed group ($\beta = -0.003$, 95% CI = [−0.06, 0.06]). However, a larger difference was detected in the low-cost condition ($\beta_{interaction} = 0.10$, 95% CI = [0.004, 0.19]). Refer to table 3 for all estimates and figure 4$d$ for illustration.

# 4. Discussion

In this registered report, we investigated whether participants with 'a cooperative phenotype' [1] (as indexed by a cooperative strategy in conditional PGG) would choose to reveal the quality of their phenotype using a costly signal and whether this signal would be associated with contributions to a group cooperative effort. Furthermore, we examined whether this relationship would hold only in the case of a highly costly signal. We found that the groups with costly signals mostly deterred participants with selfish strategies and that costlier signals were more effective (H1). However, the contrast in signal cost was not reliably estimated and needs further testing. We also found that participants with selfish intentions were more likely to mention unreasonable costs as a reason for not choosing the revealed group, and this probability increased with increasing signal cost (H2). Furthermore, groups with costly signals contributed larger portions of their remaining endowments to the common pool compared with the concealed groups, and these differences increased with increasing signal costs (H3). However, these larger contributions in the high-cost revealed group did not translate into larger earnings (H4). In summary, the results provide general support for the positive effects of costly signals on assorting cooperators and subsequent cooperation in a joint cooperative task but suggest that frequent high costs would not be evolutionarily stable. There are several important caveats to this conclusion that warrant discussion.

Whereas the separation mechanism worked rather well in the high-cost condition by driving most of the individuals playing selfishly into the concealed group and the majority of the cooperators into the revealed group, the mechanism still allowed for a semi-separating equilibrium where participants playing a selfish strategy chose the revealed group. Even more importantly, high cost deterred many cooperators from choosing the revealed group, suggesting that while increasing cost may better filter out selfish strategies, it also filters out many cooperators to the potential detriment of the joint action (as indicated by the wide 95% CIs of interaction between the group and cost variables in the statistical model testing H1). This result suggests that the mapping of cooperative phenotype on the costly signal is not a simple proportional process as envisioned by the strategic choice model [4], at least not for humans. Humans exhibit substantial communicative flexibility that can be situationally detached from the underlying phenotype. Hence, applying the strategic choice model to humans necessitates an amendment of additional parameters covering this flexibility.

One of these parameters should relate to the cost/benefit perception as suggested by Sosis [20]. We found support for his assertion that the differential perception of the cost size deters individuals playing selfishly from choosing the revealed group (H2). According to Sosis, this perception is facilitated by the socialization process whereby individuals regularly partake in costly actions (e.g. collective rituals) associated with their group's normative system, effectively discounting the perceived costs of future participation. While our data cannot attest to the role of the socialization process, we speculate that at least in our study, cooperative phenotype affected not only the perception of costs but also benefits. That is, rather than differently perceiving costs due to engaging in costly behaviours during the socialization process, it is the differential perception of potential benefits that affected the assessment of cost size as appropriate or wasteful. Looking at the pilot data—where we asked participants about their expected earnings—indicated that participants in the revealed group (compared with the concealed group) had a higher probability of saying they expect to receive the maximum amount from full cooperation ($\beta = 1.55$, 95% CI = [0.52, 1.03]). However, this finding may not apply in real life, where ritual participation does not typically precede any specific cooperative dilemma but rather a host of cooperative opportunities. In such situations, differential cost perception appears as a more likely driver of decisions to partake in costly signalling.

Notwithstanding this evidence, a stronger test of Sosis' proposition [20] would be a modification of our current design into a one-shot PGG. Although free-riders would probably invade the high-cost revealed group more often in the one-round set-up, showing that the assorting mechanism related to cost perception is functional, albeit limited, even in such a set-up would provide robust support for Sosis' proposition [20] (contra to [50]).

Regarding the effects of the separation mechanism on subsequent behaviour in PGG, we observed that participants in the high-cost revealed group invested the largest proportion of their remaining endowment (H3). Compared with participants in the low-cost revealed group, the high-cost revealed group also had lower probability of contributing nothing. Although the high-cost revealed group had the highest probability of contributing everything to the common pool, the predicted interaction effect between group and condition was not reliably estimated due to a relatively high probability of contributing everything in the high-cost concealed group, probably resulting from the large number of participants who played the cooperative strategy but refused joining the revealed group. While we did not statistically assess the difference between the revealed and concealed groups in the low-cost condition, raw data suggests that participants in the low-cost revealed group also invested larger proportions of their remaining endowment compared with participants in the concealed group. One possible explanation for this result is that despite a higher probability that a particular session in the low-cost revealed group would contain individuals playing selfishly, the low-cost signal still allowed many groups to establish a cooperative exchange. Crucially, due to a lower signal cost, the relatively successful assortment of cooperators translated into the highest earnings for this group (contra H4). This result has a plethora of interesting implications for real-life signalling contexts such as human ritual behaviour.

Since high-cost rituals often involve pain, physical effort and the expenditure of material resources, they are usually performed only on special occasions during one's lifetime (such as various rites of passage) or only occasionally during the liturgical year. In our study, the high-cost signal was sent in each PGG round, which turned out to be counterproductive in terms of the overall earnings. Having the high cost only during the first PGG round (such as an initiation ritual) or appearing only in some cyclical intervals would perhaps better simulate the real-world signalling behaviours. Indeed, for everyday mundane cooperative exchanges, low-cost regular signalling may be sufficient to stabilize a profitable level of trustworthy interactions. This conclusion is in accord with a signalling study by Chvaja *et al.* [68] that contrasted the trustworthiness of foot pilgrims to Santiago de Compostela (religious pilgrimage) with the trustworthiness resulting from participation in a Christian mass and in a secular activity. While pilgrims were rated as most trustworthy, the difference between pilgrimage and mass participation was smaller than the difference between mass participation and secular activity. Rather than a linear effect of cost, the difference between no signal and a low-cost signal is probably more important than the difference between low-cost and high-cost signal. Furthermore, in the study of two Indian villages, Power [36,38] showed that regular low-cost signals are more predictive of reputation for being trustworthy because high-cost signals may sometimes be seen as means to individual aggrandizement. While this would not be the case in our study because participants in the low-cost condition did not know about the high-cost condition, a direct comparison of high- versus low-cost choice could shed light on the perception of high-cost signals.

A preliminary inference from the current results could be that cultural evolutionary processes would pressure signal costs to be in equilibrium with expected benefits. For example, regular high-cost signals may be stable only in high-stake contexts such as combats or risky hunts where assorting cooperators without free-riders would be a crucial factor determining a group's success. In support of this conjecture, a survey of ethnographies describing ritual practices in 60 small-scale societies [54] revealed that ritual cost is positively predicted by the frequency of warfare the society experiences. Whereas the imperfect sorting mechanism in the low-cost condition afforded a profitable level of cooperative exchange in our study, the presence of free-riders would presumably disintegrate the group's cooperative effort in high-stake contexts. Our design might be easily modified to test this prediction by pitting various signalling groups against each other and comparing how the competitive context affects the workings of the sorting mechanism and subsequent cooperation.

Other important modifications to the design of the current study could alter the currency of signals and benefits. Signal costs and cooperative benefits are often disassociated in real-life settings such as costly rituals where signallers may, for example, use suffering and pain as the currency of the signal while getting helped in the future as the currency of the benefit. To provide a stronger test of CST, we decided to keep the currency of costs and benefits identical, but it could be speculated that if the cost would be, for instance, time spent on a boring task, participants in the high-cost revealed group might earn the most (due to the possibility to turn their endowment into larger investments rather

than waste them as signals). Furthermore, while having the currency of signals and benefits identical allowed us to assess the role of cooperative phenotype in signalling within the specific context of PGG, it could be speculated that cooperative phenotype would manifest differently in different cooperative contexts. To indicate this uncertainty, we talk about different cooperative strategies specific to PGG throughout the paper rather than hard-coded cooperative types [cf., [1].

In summary, this registered report provides an experimental framework that can be easily amended to examine particular extensions of CST. In our OSF repository, we provide all materials used in the current study, which can be used to replicate this study in different populations or to extend the protocol in order to further empirically develop the strategic choice model. Since cooperative communication is the cornerstone of human group living, understanding factors affecting the reliability of such communication may help us better appreciate the cooperative peculiarity of humankind.

Ethics. The research was approved by the Research Ethics Committee at Masaryk University.

Data accessibility. Data, materials and analytical code are publicly available at OSF: https://osf.io/vsjcp/. The Stage 1 manuscript associated with this Registered Report was granted in-principle acceptance on 11 June 2021 prior to data collection and analysis. The accepted Stage 1 manuscript, unchanged from the point of in-principle acceptance, may be viewed at https://osf.io/c63xk.

The data are provided in electronic supplementary material [69].

Authors' contributions. M.L.: conceptualization, data curation, formal analysis, funding acquisition, investigation, methodology, project administration, supervision, visualization, writing—original draft, writing—review and editing; R.C.: conceptualization, data curation, investigation, methodology, project administration; B.G.P.: conceptualization, methodology, supervision; D.V.: conceptualization, methodology, supervision; R.S.: conceptualization, investigation, methodology, resources, supervision.

All authors gave final approval for publication and agreed to be held accountable for the work performed therein.

Conflict of interest declaration. The authors declare no competing interests.

Funding. M.L. acknowledges the generous support from the MSCAfellow3@MUNI project [CZ.02.2.69/0.0/0.0/19_074/0012727].

Acknowledgements. We would like to thank our colleagues at the Department for the Study of Religion and Laboratory for the Experimental Study of Religion (LEVYNA) at Masaryk University, as well as colleagues at the Department for the Study of Religion at Aarhus University for helpful feedback during the development of the idea for the current study. We are grateful to Connor Wood and Dimitris Xygalatas for providing comments on an earlier draft of this registered report and Adam Kenny and two anonymous reviewers for helpful comments on Stage-1 and Stage-2 manuscripts. We further thank Jana Zuzaňáková and Katarína Čellárová for help with the preparation of the data-collection procedure, Matěj Troup for help with data collection and processing, Martin Both and Jana Zuzaňáková for preparation of the data-collection software, Theiss Bendixen for statistical advice and Markéta Poledníková for help with data processing. We would also like to thank Petra Koudelková and Dagmar Navrátilová for administrative support.

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
