## [Peer Review File · Royal Society Open Science]

Review History

RSOS-202202.R0 (Original submission)

Review form: Reviewer 1

Do you have any ethical concerns with this paper?

No

Recommendation?

Major revision

Comments to the Author(s)

This report proposes an interesting experimental investigation into the relationships between individuals' cooperative tendencies, their willingness to engage in costly signaling and collective action. The core idea is that costly religious rituals are so common across the world because they maintain cooperation by identifying and bringing together individuals committed to collective action.

Overall, this proposed study addresses a relevant research question and the motivation and methods are described in a detailed, clear and concise way. Moreover, all relevant

pilot data and code can be downloaded from OSF and I was able to reproduce their analyses and figures in a reasonable amount of time. The power analysis also seems rigorous and is presented in sufficient detail.

However, I have some major concerns about the proposed experiment and how it connects to the underlying theory, as well as some further statistical issues.

Conceptual points:

The authors set out to test an elaborate mechanism linking cooperative phenotypes, costly rituals and collective action. I do not see how the proposed design is appropriate to test any causal relationships due to a lack of experimental manipulation (or other identification strategy). In particular, the title states that it is the advertising of cooperative phenotypes through costly rituals that facilitates collective action. However, this design is unable to isolate the causal effect of costly signaling and disentangle it from any other differences between experimental conditions and pre-existing differences. It could very well be, for instance, that more cooperative individuals (as measured through self-report) behave more cooperatively and, thus, earn more in the experiment without any additional effect of costly ritual signals. To get at the causal effects, the design would require more experimental conditions and true experimental variation of costly signals (maybe also of varying strengths).

The authors call their main explanatory variable "cooperative phenotype". In contrast to the original paper (Peysakhovich Nowak & Rand, 2014), which also includes behavioral measures from multiple games, authors here rely only on self-reported measures of cooperative attitudes which might themselves be partly influenced by social desirability and value signaling. An assortment of cooperators into the signaling condition might simply arise because some individuals have a greater desire to present themselves as cooperative or "good" people, not because actual cooperators use costly signals to assort. In any case, the authors might want to use a different name to refer to self-reported cooperative values which, at best, correlate with real cooperative behavior.

The previous concern is exacerbated by the rather explicit group description which states that participants will (or will not) sacrifice part of their endowment "to demonstrate intentions regarding the size of contribution to the common pool". In order to test the prediction that costly rituals function (in an evolutionary sense) to assort cooperators, the adaptive logic should emerge endogenously from behavior and participants should not be told explicitly what the function of the costly signal might be.

I also have questions about the application of costly signaling theory to religious rituals in general. The peacock's tail only truthfully signals genetic quality because there is a shared genetic mechanism producing the signal and other (hidden) qualities. I do not see how participation in rituals is connected to other collective endeavors in such a direct way (as the authors also mention on p. 4, lines 57-59). Wouldn't individuals that only participate in collective ritual and then free-ride in actual community activities always end up with the highest payoff? Why would only truly cooperative individuals be able to participate in costly rituals? The authors mention Sosis' (2003) explanation in terms of perceived costs, but it is unclear (at least to me) how such psychological costs might affect real evolutionary dynamics that operate on material payoffs to different strategies. Recent theory suggests that costly helping behavior can evolve into an honest signal when it is adaptive for cooperators to invest in the long-term benefits of a reputation for helping

(Roberts, 2020, Honest signaling of cooperative intentions). Such reputation effects in repeated interactions might be a more plausible mechanism than perceived psychological costs.

Statistical issues:

The analysis of (transformed) PGG contributions using beta regression should be adequate, although I do not fully understand the nature of the transformation the authors apply to the raw data. It would maybe be more straightforward to analyze those data with a binomial likelihood. Say you have a total of K CZK, and G of them are given to the public pot, we have $G \sim \text{binomial}(K, p)$. In theory, this is equivalent to $G/K \sim \text{beta}(a, b)$, but the authors would not have to do any transformations and the model also automatically takes care of differences in the number of trials, i.e. total CZK. This might be relevant here as we have different (remaining) total endowments across experimental conditions.

The authors propose a three-step procedure to analyze their data, where they first introduce their main predictor (experimental group) followed by a set of covariates and, lastly, hierarchical data structure. This procedure seems rather ad-hoc and not grounded in theoretical considerations. Omitting or including predictors without explicitly considering the hypothesized causal structure among them can lead to many inferential problems (see chapter 6 in McElreath, 2020, Statistical Rethinking). The authors should justify why they think certain variables need to be controlled for in a given set of analyses. Also, hierarchical data structure -if present- should always be accounted for statistically, so there is no reason to omit random effects in the beginning. Instead, it is often helpful to first consider an "empty" model with hierarchical structure but no predictors to get an idea about how the variance is structured across different levels.

The authors plan to test H2 only in individuals choosing the concealed group. If we assume that group choice is both determined by cooperativeness and perceived costs, group choice would be a "collider" for the relationship between those variables. Using only data from the concealed group would effectively condition on the collider such that the relationship between cooperativeness and perceived costs might be biased compared to the whole population. The authors might want to consider the potential for a "collider bias" and/or run the analysis for all participants.

There are a couple of problems in the description and mathematical notation for the analytical models in table 1.

The outcome Variable Y_i in the binomial (logistic) models is whether a given individual i chose the revealed group or not (0/1). There is just 1 "trial" per participant, so $n=1$, and the goal is to model $p_i: Y_i \sim \text{binomial}(1, p_i)$. We connect the linear model to p_i using the logit link ($\text{logit}(p_i)=\dots$). Also, there is no additional error term (ϵ) in binomial models, as there is in Gaussian models. The same is true for Poisson and beta models, which also do not include an additional error.

Lastly, what does it mean that there is a single parameter for the effect of sex, age, education, and economic status? Shouldn't there be a parameter for each effect?

Minor points:

I might have missed it, but is it possible to also upload the Z-tree code to OSF?

Page 5, lines 40-42: In the revealed group, why do you expect higher individual (instead of group) payoffs? The theory predicts higher levels of cooperation that should, first and foremost, result in higher group payoffs. Individual payoffs in PGG might be maximized by free-riders who are paired with cooperators.

Pages 5-6: The authors switch between positive (e.g. "if H1-2 would be supported") and negative (e.g. "but H3-4 would not be rejected") formulations. In frequentist hypothesis testing, we can only really reject the H0, so I would suggest the authors might reformulate this section.

Page 8, line 16: Please provide some more details on how intentions are coded from participant rationales for choosing their group? What exactly is counted as "too costly"?

Review form: Reviewer 2

Do you have any ethical concerns with this paper?

No

Recommendation?

Accept with minor revision

Comments to the Author(s)

The registered report outlines a laboratory experiment designed to test whether individuals with cooperative phenotypes send costly signals, and whether such signalers subsequently benefit from assortment into groups composed of others who send costly signals. Overall, I believe the proposed study is strong, both methodologically and analytically, and will make an important contribution to key questions at the intersection of (evolutionary) anthropology and psychology. I commend the authors for detailed consideration of several aspects of the design, in particular through three pilot studies they conducted and analysed. Furthermore, in providing access to pilot data and analysis scripts, I was able to reproduce results including the power analysis.

However, the registered report could benefit from some minor revisions. These revisions mostly justify and clarify certain aspects of the proposed study, as outlined in the comments below.

The scientific validity of the research question(s).

The research questions, derived from models, are generally sound. The introduction could be broadened, as follows.

1) Not all costly signals involve large costs and/or extreme behaviour, often taking more subtle forms. Furthermore, signals might be directed at particular individuals. Could the authors comment on these aspects in the introduction? The former seems particularly relevant given that Study 2 involves a time consuming task, which could be considered a more discrete signal relative to sacrificing money. Perhaps refer readers to Bird, Ready & Power (2018) <https://doi.org/10.1038/s41562-018-0298-3>.

2) Include a statement anticipating how the results might be affected by the sample of undergraduate students from a particular European country. Furthermore, does the population either perform costly signals or participate in religious or other committed groups in ways that might affect the results, or in ways that translate to other contexts? Similarly, the abstract should mention the nationality of the sample.

3) The statement "by including reputation and experience as decision-making mechanisms" (line 28-29, pg. 7) requires further discussion. How are these mechanisms present in the proposed study?

4) Mention strategic concerns or signs of misunderstanding present in PGGs, as discussed in e.g. Burton-Chellew, El Mouden & West (2017)

<https://doi.org/10.1098/rspb.2017.0689>. Strategic concerns might be particularly pertinent as "ritual sacrifice" (as per the proposed study) could be considered not as a signal of hidden quality but as a method to induce cooperation in others. Additionally, these motivations could be assessed in the written rationale provided by participants.

The logic, rationale, and plausibility of the proposed hypotheses.

Generally, the proposed hypotheses are clear and follow from theory. However, the following could be clarified in relation to the hypotheses.

5) The four hypotheses in Table 1 are clear and specific (although see comment 6). Because they relate to the experimental design they are more accurate than the initial verbal description of the hypotheses. I think the terms "ritual sacrifice" and "group norms" in the verbal description can be linked more explicitly to the experiment. My questions surrounding these two terms are:

a) ritual sacrifice: it is clear that the one-time decision to discard part of the endowment represents a sacrifice. Is the ritual element represented by the sacrifice occurring before each of the three rounds?

b) group norms: it is unclear how choosing to cooperate in the revealed group represents adherence to cooperative norms. The revealed and the concealed group both include the instruction "it is expected that every member should contribute as much as possible". Thus there is no priming difference. The only difference in expectations would come from participants self-selecting into groups, with the expectation that others will e.g. cooperate in the revealed group. Even so, there is no established norm as such, at least not in the first round of the PGG. Could the authors justify the use of the term "norms"?

6) Table 1, hypotheses 3 and 4 require revision. Currently, H3 and H4 refer to endowment and earnings in the "second PGG scenario" --- should H3 and H4 relate to either one or all of the three rounds in the PGG? If the authors plan to analyse the proportion contributed per round, should a random effect for the intercepts of each participant be included?

7) It is unclear how the authors will investigate signal cost and between-group conflict (Table 1, interpretations) --- could the authors elaborate?

The soundness and feasibility of the methodology and analysis pipeline (including statistical power analysis where applicable).

Generally, the methods including analysis are sound and feasible. I was able to run the analyses for the pilot studies. The analysis pipeline for the proposed is sound (although

check comment 6). I was able to run the power analysis (for the unreported H1 and H2, as well as the reported H4). The study is sufficiently powered to detect a small effect size.

8) Methods: Assessment would have been easier with access to the materials (original and/or translated) for the proposed studies. For example, I was unable to assess the comprehension questions. Could the authors provide materials?

9) Methods: Would it be worth asking individuals about religion and religious participation? These measures could be additional covariates included in the second step for each analysis. Alternatively, these covariates could be included as descriptive sample characteristics, to get a sense of whether or not this sample engages in rituals that the experiment is meant to represent. I leave it to the authors to decide whether the inclusion of such variables is warranted!

10) Method (minor, pg. 7): Make clear that the sacrificed 10% is discarded and not be available during the PGG.

Whether the clarity and degree of methodological detail would be sufficient to replicate the proposed experimental procedures and analysis pipeline.

Generally, the methods including analysis pipeline are clear and sufficiently detailed. However, the following would add further detail to guarantee replication.

11) Is it the case that more than 104 participants will take part in the online questionnaire (per study), but 104 will be selected to take part (discounting the extra participant invited to each session)? If so, this should be made clear, indicating the approximate number of participants expected to be recruited to complete the questionnaire.

12) Pilot 3 should be briefly mentioned under Pilot Data in the manuscript, to justify the interval of the time-consuming task.

13) Minor: correct reference to figure 1E (SM, pg. 17).

Whether the authors provide a sufficiently clear and detailed description of the methods to prevent undisclosed flexibility in the experimental procedures or analysis pipeline.

Again, the methods including analysis pipeline are clear and sufficiently detailed. However, the following would help prevent flexibility in procedures and analysis.

14) The authors should provide more detail (possibly in the SM) of how they will analyse the written rationales indicated by participants for choosing their group. Currently, they report that they will look for mentions of high costs. Could they provide a series of key terms/phrases they will use during the coding of responses? Could they provide some examples of what this looked like in the pilot, using original and translated written rationales? Eventually, in the final analysis, other researchers should be able to independently reproduce the coding of rationales, so the authors should consider sharing original and/or translated rationales.

15) Given that in the pilot a participant was excluded for indicating an unlikely age, it is perhaps worth stating what will be done with unlikely data (e.g. excluded or treated as missing data and using the proposed multiple imputation method). It is not necessary to

indicate potential unlikely data for each variable, but signalling that this might be an issue might be worthwhile given it occurred with the pilot. Alternatively, the online questionnaire could be designed in such a way to avoid this issue.

16) Can the authors confirm that the trustworthiness scale will not be included in the proposed analysis? If so, the authors should justify why the trustworthiness scale is being included in the proposed study.

17) Demographic variables such as education and economic status are included in the analyses. Do they authors have any expectations regarding these variables? Furthermore, could the authors justify the specific economic status question used?

18) Define "substantial effect" (Table 1, interpretations column).

Whether the authors have considered sufficient outcome-neutral conditions (e.g. absence of floor or ceiling effects; positive controls; other quality checks) for ensuring that the results obtained are able to test the stated hypotheses.

The authors have considered sufficient outcome-neutral conditions, and the pilot data suggest there will be no floor or ceiling effects. They have also considered quality checks during the pilot so that the experiment should overcome certain potential issues.

Review form: Reviewer 3

Do you have any ethical concerns with this paper?

No

Recommendation?

Major revision

Comments to the Author(s)

This registered report describes the development of a new paradigm which aims to study the effects of 'costly ritual signals on collective action' in a controlled laboratory setting. The manuscript describes two studies designed to test multiple hypotheses derived from theories of costly signaling and their applications to the domain of religion: (H1) that individuals with a 'cooperative phenotype' will be more likely to sacrifice some of their resources to signal their commitment to cooperating in a social dilemma; (H2) relatedly, that individuals without this 'cooperative phenotype' will be less likely to send this signal because they perceive higher costs of doing so; (H3) that individuals who choose to send a costly signal will cooperate more than those who don't; and (H4) that groups composed of signalers will achieve higher earnings than groups composed of non-signalers.

To test these hypotheses, the authors propose two studies in which they will first measure participants' 'cooperative phenotype' via self-report measures of cooperative values and trustworthiness; then ask participants to choose between interacting either in a 'revealed' group, composed of individuals who choose to signal their cooperative intentions or in a 'concealed' group, composed of individuals who choose not to signal their cooperative intentions; and finally have participants interact repeatedly (for three periods) in a public goods game. In Study 1, signaling will be costly because participants will have to spend

10% of their initial endowment to signal; in Study 2, signaling will be costly because participants will need to spend 24 minutes completing a boring task to be able to signal.

In my view, this research sets out to test interesting questions about the role of ritualized costly displays in facilitating coordination and helping to solve collective action problems. However, I have some doubts regarding the extent to which the proposed experimental design can help address such questions. To me, it seems that the motivating question for this work is why people engage in extreme ritualized behaviors, which often involve self-harm and very costly sacrifices. I think the proposed paradigm might not be ideal to answer this question, for two main reasons: (a) because the costs involved in the aforementioned ritualized behaviors seem extremely high (and this is exactly what makes these behaviors so puzzling), whereas the costs paid by participants to signal in the proposed studies are very low by comparison; (b) because the behaviors of interest are culturally prescribed and, as the authors note, are based on long-term socialization processes and internalized norms, whereas signaling behaviors in the proposed studies lack cultural meaning and do not appear ritualized. Another related concern is that (c) many ritual performances seem to signal norm abidance rather than cooperative intentions per se, but in this study the only 'norm' participants are exposed to (and can signal abidance to) is a norm of cooperation. (A more stringent test of the idea that ritualized behaviors per se signal an underlying cooperative phenotype, would be to examine the effects of costly following of norms that are not explicitly related to cooperation. Would individuals that signal abidance to any group norm contribute more to collective action?)

To sum up, I think the proposed studies are interesting and they can answer important questions regarding who is willing to pay costs to signal cooperative intentions; whether such signals are honest; and how assorting in groups of costly signalers affects cooperation and group payoffs. However, I think that the experimental design does not capture important aspects of costly ritual displays, including their extremely high costs, cultural significance, and potential to signal abidance to any group norms – whether cooperative or not.

With regards to the methodology, I think that the design of the two studies is sound, feasible, and described (with few exceptions) in enough detail to allow replicating the experimental procedures. A couple of exceptions: information on the trustworthiness scale is lacking as is a precise description of the cooperation measure. More specifically, it would be helpful if the authors include the items of the trustworthiness scale they used in the Supplemental Materials. Further, if they decide to use this measure in their main studies, they should report all results when using trustworthiness as an outcome measure (in addition to results with the cooperative values scale as the outcome measure). With regards to the measure of cooperation, how exactly did participants make their decisions, e.g., could they choose to contribute any percentage of their endowment or did they have specific options such as contributing in increments of 10?

Finally, besides these methodological details, I have two broader comments regarding the study design. The first concerns the measures used to assess the 'cooperative phenotype.' While I see that these measures have been shown to correlate with cooperative behavior in experimental games, I am unsure about the extent to which they have been properly validated. As the authors note based on their pilot data, not all items of the cooperative values scale seem to tap a common underlying factor; further, some of the items on that scale could be measuring participants' political orientation, besides their general

cooperative tendencies. Alternative to the cooperative values and trustworthiness questionnaires, the authors could consider using well-validated measures of Social Value Orientations (e.g., the SVO Slider scale; Murphy et al., 2011). This would have the added benefit of allowing them to distinguish between joint-gain maximizers, altruists, and individuals who are inequality averse in their sample. Such distinctions may be relevant to the interpretation of results as, at least in the pilot, some high-contributing participants seem to choose the 'concealed' group to avoid wasting resources that can otherwise benefit the collective.

My second comment concerns the number of rounds of the PGG. I see how including more rounds can allow a better understanding of cooperation dynamics in the 'revealed' and 'concealed' groups. It is reasonable to expect that cooperation can be better maintained in the 'revealed' group, but that it will break down over time in the 'concealed' group. However, I am not sure that three rounds are enough to observe this pattern and would suggest that the authors consider studying interactions over a larger number of rounds (see Gächter et al., 2008, for an example of comparing shorter-term versus longer-term dynamics). Finally, one thing to consider is that, over time, real-life groups might need fewer and fewer costly signals to maintain coordination and cooperation. This would mean that, over time, group payoffs would increase in groups of signalers, as they would achieve cooperation with fewer and fewer costs. It seems that the design of the current study precludes studying this kind of process, because participants make a one-off decision to signal or not across all PGG rounds.

References

- Gächter, S., Renner, E., & Sefton, M. (2008). The long-run benefits of punishment. *Science*, 322(5907), 1510-1510.
- Murphy, R. O., Ackermann, K. A., & Handgraaf, M. (2011). Measuring social value orientation. *Judgment and Decision Making*, 6(8), 771-781.

Decision letter (RSOS-202202.R0)

Dear Mr Lang,

The Editors assigned to your stage one Registered Report ("Advertising cooperative phenotype through costly ritual signals facilitates collective action") have now received comments from reviewers. We would like you to revise your paper in accordance with the referee and editors suggestions which can be found below (not including confidential reports to the Editor). Please note this decision does not guarantee eventual acceptance.

Please submit a copy of your revised paper within six weeks (i.e. by 31 March 2021). If we do not hear from you within this time then it will be assumed that the paper has been withdrawn. If deemed necessary by the Editors, your manuscript will be sent back to one

or more of the original reviewers for assessment. If the original reviewers are not available we may invite new reviewers.

When submitting your revised manuscript, you must respond to the comments made by the referees and upload a file "Response to Referees" in "Section 2 - File Upload". Please use this to document how you have responded to the comments, and the adjustments you have made. In order to expedite the processing of the revised manuscript, please be as specific as possible in your response.

Additionally, when resubmitting, please ensure that the emails of all co-authors are up to date. At present, the following email is showing as incorrect:

radim.chvaja@mail.muni.cz

on behalf of Professor Chris Chambers (Registered Reports Editor, Royal Society Open Science) openscience@royalsociety.org

Associate Editor Comments to Author (Professor Chris Chambers):

Thank you for your patience during this challenging time for reviewers. Three expert reviewers have now assessed the manuscript and all find a degree of promise in the proposal while also highlighting a considerable breadth and depth of issues that will need to be addressed to achieved Stage 1 in-principle acceptance. Major shortcomings to address include the strength of the theoretical rationale, suitability of the methodology to answer the research question, statistical concerns (including potential collider bias), clarity of the predictions, limitations of the sampling plan, and the level of methodological detail. Several of these concerns are shared across reviewers.

Were the study already completed and in receipt these reviews, the editorial decision would be to reject, but the advantage of the Registered Reports process is that Stage 1 assessment provides the opportunity to optimise and adjust study designs in order to address and circumvent problems that would otherwise become irreversible. In this case, although the issues raised are significant, the reviews are sufficiently detailed, constructive and enthusiastic that I would like to invite the authors to submit a major

revision. Please be sure to carefully address every issue raised by the reviewers in a revised manuscript and response.

Comments to Author:

Reviewer: 1

Comments to the Author(s)

This report proposes an interesting experimental investigation into the relationships between individuals' cooperative tendencies, their willingness to engage in costly signaling and collective action. The core idea is that costly religious rituals are so common across the world because they maintain cooperation by identifying and bringing together individuals committed to collective action.

Overall, this proposed study addresses a relevant research question and the motivation and methods are described in a detailed, clear and concise way. Moreover, all relevant pilot data and code can be downloaded from OSF and I was able to reproduce their analyses and figures in a reasonable amount of time. The power analysis also seems rigorous and is presented in sufficient detail.

However, I have some major concerns about the proposed experiment and how it connects to the underlying theory, as well as some further statistical issues.

Conceptual points:

The authors set out to test an elaborate mechanism linking cooperative phenotypes, costly rituals and collective action. I do not see how the proposed design is appropriate to test any causal relationships due to a lack of experimental manipulation (or other identification strategy). In particular, the title states that it is the advertising of cooperative phenotypes through costly rituals that facilitates collective action. However, this design is unable to isolate the causal effect of costly signaling and disentangle it from any other differences between experimental conditions and pre-existing differences. It could very well be, for instance, that more cooperative individuals (as measured through self-report) behave more cooperatively and, thus, earn more in the experiment without any additional effect of costly ritual signals. To get at the causal effects, the design would require more experimental conditions and true experimental variation of costly signals (maybe also of varying strengths).

The authors call their main explanatory variable "cooperative phenotype". In contrast to the original paper (Peysakhovich Nowak & Rand, 2014), which also includes behavioral measures from multiple games, authors here rely only on self-reported measures of cooperative attitudes which might themselves be partly influenced by social desirability and value signaling. An assortment of cooperators into the signaling condition might simply arise because some individuals have a greater desire to present themselves as cooperative or "good" people, not because actual cooperators use costly signals to assort. In any case, the authors might want to use a different name to refer to self-reported cooperative values which, at best, correlate with real cooperative behavior.

The previous concern is exacerbated by the rather explicit group description which states that participants will (or will not) sacrifice part of their endowment "to demonstrate intentions regarding the size of contribution to the common pool". In order to test the prediction that costly rituals function (in an evolutionary sense) to assort cooperators, the

adaptive logic should emerge endogenously from behavior and participants should not be told explicitly what the function of the costly signal might be.

I also have questions about the application of costly signaling theory to religious rituals in general. The peacock's tail only truthfully signals genetic quality because there is a shared genetic mechanism producing the signal and other (hidden) qualities. I do not see how participation in rituals is connected to other collective endeavors in such a direct way (as the authors also mention on p. 4, lines 57-59). Wouldn't individuals that only participate in collective ritual and then free-ride in actual community activities always end up with the highest payoff? Why would only truly cooperative individuals be able to participate in costly rituals? The authors mention Sosis' (2003) explanation in terms of perceived costs, but it is unclear (at least to me) how such psychological costs might affect real evolutionary dynamics that operate on material payoffs to different strategies. Recent theory suggests that costly helping behavior can evolve into an honest signal when it is adaptive for cooperators to invest in the long-term benefits of a reputation for helping (Roberts, 2020, Honest signaling of cooperative intentions). Such reputation effects in repeated interactions might be a more plausible mechanism than perceived psychological costs.

Statistical issues:

The analysis of (transformed) PGG contributions using beta regression should be adequate, although I do not fully understand the nature of the transformation the authors apply to the raw data. It would maybe be more straightforward to analyze those data with a binomial likelihood. Say you have a total of K CZK, and G of them are given to the public pot, we have $G \sim \text{binomial}(K, p)$. In theory, this is equivalent to $G/K \sim \text{beta}(a, b)$, but the authors would not have to do any transformations and the model also automatically takes care of differences in the number of trials, i.e. total CZK. This might be relevant here as we have different (remaining) total endowments across experimental conditions.

The authors propose a three-step procedure to analyze their data, where they first introduce their main predictor (experimental group) followed by a set of covariates and, lastly, hierarchical data structure. This procedure seems rather ad-hoc and not grounded in theoretical considerations. Omitting or including predictors without explicitly considering the hypothesized causal structure among them can lead to many inferential problems (see chapter 6 in McElreath, 2020, *Statistical Rethinking*). The authors should justify why they think certain variables need to be controlled for in a given set of analyses. Also, hierarchical data structure -if present- should always be accounted for statistically, so there is no reason to omit random effects in the beginning. Instead, it is often helpful to first consider an "empty" model with hierarchical structure but no predictors to get an idea about how the variance is structured across different levels.

The authors plan to test H2 only in individuals choosing the concealed group. If we assume that group choice is both determined by cooperativeness and perceived costs, group choice would be a "collider" for the relationship between those variables. Using only data from the concealed group would effectively condition on the collider such that the relationship between cooperativeness and perceived costs might be biased compared to the whole population. The authors might want to consider the potential for a "collider bias" and/or run the analysis for all participants.

There are a couple of problems in the description and mathematical notation for the analytical models in table 1.

The outcome Variable Y_i in the binomial (logistic) models is whether a given individual i chose the revealed group or not (0/1). There is just 1 "trial" per participant, so $n=1$, and the goal is to model $p_i: Y_i \sim \text{binomial}(1, p_i)$. We connect the linear model to p_i using the logit link ($\text{logit}(p_i)=\dots$). Also, there is no additional error term (epsilon) in binomial models, as there is in Gaussian models. The same is true for Poisson and beta models, which also do not include an additional error.

Lastly, what does it mean that there is a single parameter for the effect of sex, age, education, and economic status? Shouldn't there be a parameter for each effect?

Minor points:

I might have missed it, but is it possible to also upload the Z-tree code to OSF?

Page 5, lines 40-42: In the revealed group, why do you expect higher individual (instead of group) payoffs? The theory predicts higher levels of cooperation that should, first and foremost, result in higher group payoffs. Individual payoffs in PGG might be maximized by free-riders who are paired with cooperators.

Pages 5-6: The authors switch between positive (e.g. "if H1-2 would be supported") and negative (e.g. "but H3-4 would not be rejected") formulations. In frequentist hypothesis testing, we can only really reject the H_0 , so I would suggest the authors might reformulate this section.

Page 8, line 16: Please provide some more details on how intentions are coded from participant rationales for choosing their group? What exactly is counted as "too costly"?

Reviewer: 2

Comments to the Author(s)

The registered report outlines a laboratory experiment designed to test whether individuals with cooperative phenotypes send costly signals, and whether such signalers subsequently benefit from assortment into groups composed of others who send costly signals. Overall, I believe the proposed study is strong, both methodologically and analytically, and will make an important contribution to key questions at the intersection of (evolutionary) anthropology and psychology. I commend the authors for detailed consideration of several aspects of the design, in particular through three pilot studies they conducted and analysed. Furthermore, in providing access to pilot data and analysis scripts, I was able to reproduce results including the power analysis.

However, the registered report could benefit from some minor revisions. These revisions mostly justify and clarify certain aspects of the proposed study, as outlined in the comments below.

The scientific validity of the research question(s).

The research questions, derived from models, are generally sound. The introduction could be broadened, as follows.

1) Not all costly signals involve large costs and/or extreme behaviour, often taking more subtle forms. Furthermore, signals might be directed at particular individuals. Could the authors comment on these aspects in the introduction? The former seems particularly relevant given that Study 2 involves a time consuming task, which could be considered a more discrete signal relative to sacrificing money. Perhaps refer readers to Bird, Ready & Power (2018) <https://doi.org/10.1038/s41562-018-0298-3>.

2) Include a statement anticipating how the results might be affected by the sample of undergraduate students from a particular European country. Furthermore, does the population either perform costly signals or participate in religious or other committed groups in ways that might affect the results, or in ways that translate to other contexts? Similarly, the abstract should mention the nationality of the sample.

3) The statement "by including reputation and experience as decision-making mechanisms" (line 28-29, pg. 7) requires further discussion. How are these mechanisms present in the proposed study?

4) Mention strategic concerns or signs of misunderstanding present in PGGs, as discussed in e.g. Burton-Chellew, El Mouden & West (2017) <https://doi.org/10.1098/rspb.2017.0689>. Strategic concerns might be particularly pertinent as "ritual sacrifice" (as per the proposed study) could be considered not as a signal of hidden quality but as a method to induce cooperation in others. Additionally, these motivations could be assessed in the written rationale provided by participants.

The logic, rationale, and plausibility of the proposed hypotheses.

Generally, the proposed hypotheses are clear and follow from theory. However, the following could be clarified in relation to the hypotheses.

5) The four hypotheses in Table 1 are clear and specific (although see comment 6). Because they relate to the experimental design they are more accurate than the initial verbal description of the hypotheses. I think the terms "ritual sacrifice" and "group norms" in the verbal description can be linked more explicitly to the experiment. My questions surrounding these two terms are:

a) ritual sacrifice: it is clear that the one-time decision to discard part of the endowment represents a sacrifice. Is the ritual element represented by the sacrifice occurring before each of the three rounds?

b) group norms: it is unclear how choosing to cooperate in the revealed group represents adherence to cooperative norms. The revealed and the concealed group both include the instruction "it is expected that every member should contribute as much as possible". Thus there is no priming difference. The only difference in expectations would come from participants self-selecting into groups, with the expectation that others will e.g. cooperate in the revealed group. Even so, there is no established norm as such, at least not in the first round of the PGG. Could the authors justify the use of the term "norms"?

6) Table 1, hypotheses 3 and 4 require revision. Currently, H3 and H4 refer to endowment and earnings in the "second PGG scenario" --- should H3 and H4 relate to either one or all of the three rounds in the PGG? If the authors plan to analyse the proportion contributed per round, should a random effect for the intercepts of each participant be included?

7) It is unclear how the authors will investigate signal cost and between-group conflict (Table 1, interpretations) --- could the authors elaborate?

The soundness and feasibility of the methodology and analysis pipeline (including statistical power analysis where applicable).

Generally, the methods including analysis are sound and feasible. I was able to run the analyses for the pilot studies. The analysis pipeline for the proposed is sound (although check comment 6). I was able to run the power analysis (for the unreported H1 and H2, as well as the reported H4). The study is sufficiently powered to detect a small effect size.

8) Methods: Assessment would have been easier with access to the materials (original and/or translated) for the proposed studies. For example, I was unable to assess the comprehension questions. Could the authors provide materials?

9) Methods: Would it be worth asking individuals about religion and religious participation? These measures could be additional covariates included in the second step for each analysis. Alternatively, these covariates could be included as descriptive sample characteristics, to get a sense of whether or not this sample engages in rituals that the experiment is meant to represent. I leave it to the authors to decide whether the inclusion of such variables is warranted!

10) Method (minor, pg. 7): Make clear that the sacrificed 10% is discarded and not be available during the PGG.

Whether the clarity and degree of methodological detail would be sufficient to replicate the proposed experimental procedures and analysis pipeline.

Generally, the methods including analysis pipeline are clear and sufficiently detailed. However, the following would add further detail to guarantee replication.

11) Is it the case that more than 104 participants will take part in the online questionnaire (per study), but 104 will be selected to take part (discounting the extra participant invited to each session)? If so, this should be made clear, indicating the approximate number of participants expected to be recruited to complete the questionnaire.

12) Pilot 3 should be briefly mentioned under Pilot Data in the manuscript, to justify the interval of the time-consuming task.

13) Minor: correct reference to figure 1E (SM, pg. 17).

Whether the authors provide a sufficiently clear and detailed description of the methods to prevent undisclosed flexibility in the experimental procedures or analysis pipeline.

Again, the methods including analysis pipeline are clear and sufficiently detailed. However, the following would help prevent flexibility in procedures and analysis.

14) The authors should provide more detail (possibly in the SM) of how they will analyse the written rationales indicated by participants for choosing their group. Currently, they report that they will look for mentions of high costs. Could they provide a series of key terms/phrases they will use during the coding of responses? Could they provide some

examples of what this looked like in the pilot, using original and translated written rationales? Eventually, in the final analysis, other researchers should be able to independently reproduce the coding of rationales, so the authors should consider sharing original and/or translated rationales.

15) Given that in the pilot a participant was excluded for indicating an unlikely age, it is perhaps worth stating what will be done with unlikely data (e.g. excluded or treated as missing data and using the proposed multiple imputation method). It is not necessary to indicate potential unlikely data for each variable, but signalling that this might be an issue might be worthwhile given it occurred with the pilot. Alternatively, the online questionnaire could be designed in such a way to avoid this issue.

16) Can the authors confirm that the trustworthiness scale will not be included in the proposed analysis? If so, the authors should justify why the trustworthiness scale is being included in the proposed study.

17) Demographic variables such as education and economic status are included in the analyses. Do they authors have any expectations regarding these variables? Furthermore, could the authors justify the specific economic status question used?

18) Define "substantial effect" (Table 1, interpretations column).

Whether the authors have considered sufficient outcome-neutral conditions (e.g. absence of floor or ceiling effects; positive controls; other quality checks) for ensuring that the results obtained are able to test the stated hypotheses.

The authors have considered sufficient outcome-neutral conditions, and the pilot data suggest there will be no floor or ceiling effects. They have also considered quality checks during the pilot so that the experiment should overcome certain potential issues.

Reviewer: 3

Comments to the Author(s)

This registered report describes the development of a new paradigm which aims to study the effects of 'costly ritual signals on collective action' in a controlled laboratory setting. The manuscript describes two studies designed to test multiple hypotheses derived from theories of costly signaling and their applications to the domain of religion: (H1) that individuals with a 'cooperative phenotype' will be more likely to sacrifice some of their resources to signal their commitment to cooperating in a social dilemma; (H2) relatedly, that individuals without this 'cooperative phenotype' will be less likely to send this signal because they perceive higher costs of doing so; (H3) that individuals who choose to send a costly signal will cooperate more than those who don't; and (H4) that groups composed of signalers will achieve higher earnings than groups composed of non-signalers.

To test these hypotheses, the authors propose two studies in which they will first measure participants' 'cooperative phenotype' via self-report measures of cooperative values and trustworthiness; then ask participants to choose between interacting either in a 'revealed' group, composed of individuals who choose to signal their cooperative intentions or in a 'concealed' group, composed of individuals who choose not to signal their cooperative intentions; and finally have participants interact repeatedly (for three periods) in a public goods game. In Study 1, signaling will be costly because participants will have to spend

10% of their initial endowment to signal; in Study 2, signaling will be costly because participants will need to spend 24 minutes completing a boring task to be able to signal.

In my view, this research sets out to test interesting questions about the role of ritualized costly displays in facilitating coordination and helping to solve collective action problems. However, I have some doubts regarding the extent to which the proposed experimental design can help address such questions. To me, it seems that the motivating question for this work is why people engage in extreme ritualized behaviors, which often involve self-harm and very costly sacrifices. I think the proposed paradigm might not be ideal to answer this question, for two main reasons: (a) because the costs involved in the aforementioned ritualized behaviors seem extremely high (and this is exactly what makes these behaviors so puzzling), whereas the costs paid by participants to signal in the proposed studies are very low by comparison; (b) because the behaviors of interest are culturally prescribed and, as the authors note, are based on long-term socialization processes and internalized norms, whereas signaling behaviors in the proposed studies lack cultural meaning and do not appear ritualized. Another related concern is that (c) many ritual performances seem to signal norm abidance rather than cooperative intentions per se, but in this study the only 'norm' participants are exposed to (and can signal abidance to) is a norm of cooperation. (A more stringent test of the idea that ritualized behaviors per se signal an underlying cooperative phenotype, would be to examine the effects of costly following of norms that are not explicitly related to cooperation. Would individuals that signal abidance to any group norm contribute more to collective action?)

To sum up, I think the proposed studies are interesting and they can answer important questions regarding who is willing to pay costs to signal cooperative intentions; whether such signals are honest; and how assorting in groups of costly signalers affects cooperation and group payoffs. However, I think that the experimental design does not capture important aspects of costly ritual displays, including their extremely high costs, cultural significance, and potential to signal abidance to any group norms – whether cooperative or not.

With regards to the methodology, I think that the design of the two studies is sound, feasible, and described (with few exceptions) in enough detail to allow replicating the experimental procedures. A couple of exceptions: information on the trustworthiness scale is lacking as is a precise description of the cooperation measure. More specifically, it would be helpful if the authors include the items of the trustworthiness scale they used in the Supplemental Materials. Further, if they decide to use this measure in their main studies, they should report all results when using trustworthiness as an outcome measure (in addition to results with the cooperative values scale as the outcome measure). With regards to the measure of cooperation, how exactly did participants make their decisions, e.g., could they choose to contribute any percentage of their endowment or did they have specific options such as contributing in increments of 10?

Finally, besides these methodological details, I have two broader comments regarding the study design. The first concerns the measures used to assess the 'cooperative phenotype.' While I see that these measures have been shown to correlate with cooperative behavior in experimental games, I am unsure about the extent to which they have been properly validated. As the authors note based on their pilot data, not all items of the cooperative values scale seem to tap a common underlying factor; further, some of the items on that scale could be measuring participants' political orientation, besides their general

cooperative tendencies. Alternative to the cooperative values and trustworthiness questionnaires, the authors could consider using well-validated measures of Social Value Orientations (e.g., the SVO Slider scale; Murphy et al., 2011). This would have the added benefit of allowing them to distinguish between joint-gain maximizers, altruists, and individuals who are inequality averse in their sample. Such distinctions may be relevant to the interpretation of results as, at least in the pilot, some high-contributing participants seem to choose the 'concealed' group to avoid wasting resources that can otherwise benefit the collective.

My second comment concerns the number of rounds of the PGG. I see how including more rounds can allow a better understanding of cooperation dynamics in the 'revealed' and 'concealed' groups. It is reasonable to expect that cooperation can be better maintained in the 'revealed' group, but that it will break down over time in the 'concealed' group. However, I am not sure that three rounds are enough to observe this pattern and would suggest that the authors consider studying interactions over a larger number of rounds (see Gächter et al., 2008, for an example of comparing shorter-term versus longer-term dynamics). Finally, one thing to consider is that, over time, real-life groups might need fewer and fewer costly signals to maintain coordination and cooperation. This would mean that, over time, group payoffs would increase in groups of signalers, as they would achieve cooperation with fewer and fewer costs. It seems that the design of the current study precludes studying this kind of process, because participants make a one-off decision to signal or not across all PGG rounds.

References

Gächter, S., Renner, E., & Sefton, M. (2008). The long-run benefits of punishment. *Science*, 322(5907), 1510-1510.

Murphy, R. O., Ackermann, K. A., & Handgraaf, M. (2011). Measuring social value orientation. *Judgment and Decision Making*, 6(8), 771-781.

Author's Response to Decision Letter for (RSOS-202202.R0)

See Appendix A.

RSOS-202202.R1 (Revision)

Review form: Reviewer 1

Do you have any ethical concerns with this paper?

No

Recommendation?

Accept with minor revision

Comments to the Author(s)

The authors present a revised and much improved version of their registered report proposing an experimental investigation into cooperative tendencies, costly (ritualistic) signalling and collective action.

The biggest concern I had with the first version of this manuscript was the lack of experimental manipulation and the impossibility to detect the independent causal effect of costly signals in the assortment of cooperators. To remedy this problem the authors now distinguish between low-cost and high-cost conditions and randomly assign participants to those conditions. So participants either choose between concealed and low-cost revealed or between concealed and high-cost revealed. The authors hypothesize that only high cost would function to assort cooperators. This manipulation should allow authors to better disentangle the effect of costly signals from other group differences and ameliorate the limitations of self-selection into experimental conditions.

The authors also clarified the theoretical rationale of their approach and included recent modeling work on costly signalling and reputation in repeated interactions (Roberts, 2020). This makes the introduction both clearer and better grounded in formal theory.

The authors have decided to include a behavioral measure of "cooperative phenotype" based on hypothetical PGG contributions instead of the self-report measure they used before. In general, I agree that behavioral measures are often more predictive of (real-world) behavior in other contexts, but I worry about the independence of this measure as an explanatory variable. It might not be surprising that individuals who indicate cooperative intentions in a hypothetical PGG will also behave more cooperatively in a real PGG. Furthermore, participants might even remember the answer given in the hypothetical scenario and might play a similar strategy in the actual game independently of their actual latent cooperative dispositions. The authors might want to comment on this potential issue.

The authors have improved their statistical notation, but there are still some issues. The link function in GLMs is not applied to the outcome but to a parameter in the likelihood. Focussing on the binomial model for H1: The outcome Y_i (0 or 1, not the probability as stated by the authors) is binomially distributed, so $Y_i \sim \text{binom}(1, p_i)$, "n=1" because there is just one "trial" per participant. We then connect the participant-specific probability p_i to the linear model as $\text{logit}(p_i) = \dots$. In my view, including the model equations is also not absolutely necessary as long as the code is accessible and the models are correctly described, but if authors decide to include equations, they should make sure they are correct.

All remaining points were addressed satisfactorily by the authors. Thank you for your effort and rigor!

Review form: Reviewer 2

Do you have any ethical concerns with this paper?

No

Recommendation?

Accept in principle

Comments to the Author(s)

The authors have considered the comments made by me and the other two reviewers; I'm satisfied with how my comments have been addressed. Considering all comments it appears that the authors have incorporated suggestions in most instances. In cases where they have not, the authors have provided a sound justification for not including changes at this time. For example, I think it's appropriate that they plan to provide possible extensions in the eventual discussion, because the proposed study is an initial attempt to establish the relationship between cooperative phenotype, costly signalling, and behaviour. I think that the three major methodological changes --- inclusion of the Fischbacher et al. (2001) behavioural assessment, replacement of an effortful condition with a low-cost condition, and more public goods game iterations --- have improved the proposed study. Some of these changes are in line with adjustment of the general framing in the introduction, which is also clear.

Two notes, neither of which prevent the manuscript being accepted. 1) Could Figure S1 be moved to the manuscript? I find it a useful reference! 2) Could it be specified in H3, Table 1 that a "future" or "follow-up" study would investigate the effect of between-group competition? This would make clear that such a design would be warranted as part of a new study if the results of the current one are not confirmed (as the authors intend) and is not necessary as part of the current study. This future work could potentially consider other explanations, as group competition might be one of several plausible contexts under which the effects of costly signals may be observed.

I look forward to seeing the results of this high quality study, which should stimulate insights and further research into the role of costly signalling in cooperation.

Review form: Reviewer 3

Do you have any ethical concerns with this paper?

No

Recommendation?

Accept with minor revision

Comments to the Author(s)

Thank you for inviting me to review this revised RR. My overall impression is that the manuscript has substantially improved and I am for the most part satisfied with the authors' responses to my previous comments. Below, I am describing a few remaining concerns:

1. I think that manipulating norms in a laboratory setting would be feasible, without adverse real-life consequences for participants (although mimicking real-life socialization processes would be very challenging in the lab). That said, I appreciate that the authors have re-written the introduction in a way that emphasizes cooperative phenotype and cooperative intention signaling, and de-emphasizes participation in culturally specific and extremely costly rituals. I think the current version of the introduction fits more closely with the questions that the proposed study can investigate.

2. In the revision, the authors suggest that it is not only extreme ritual behaviors that can signal a cooperative phenotype, but that more mundane behaviors, such as church attendance, can work in a similar way. This gives rise to the question of how costly a behavior needs to be in order to act as an honest signal of an underlying cooperative phenotype. I think that the authors' decision to manipulate the cost (high versus low) of signaling is a great addition to the design. But, it is possible with this design that the authors find support for their hypotheses, albeit not only for the high-cost condition. What would the authors conclude if, for example, participants with a cooperative phenotype are equally likely to signal in both the high- and low-cost conditions? Or if both low-cost and high-cost signaling are sufficient to boost contributions and stabilize cooperation? Would this go against the broader theory, or would it suggest that relatively low-cost signals might still allow assortment of cooperators (and maintenance of contributions at high levels)?
3. I think that the addition of the behavioral task to measure participants' cooperative phenotype is a significant improvement over the previous measures.
4. I have one last clarification question concerning the design: What feedback will participants receive between rounds of the PGG? In the manuscript, the authors write that "Using their remaining endowment, participants in both groups will make first iteration PGG decisions and learn about their current earnings." Does this mean that participants will not receive information about other group members' contributions? If so, this might allow selfish individuals to 'fake' a cooperative phenotype by signaling, but keep making low contributions while remaining undetected (e.g., if they happen to be in a revealed group in which all other group members are high contributors).

Decision letter (RSOS-202202.R1)

Dear Mr Lang,

On behalf of the Editors, I am pleased to inform you that your Manuscript RSOS-202202.R1 entitled "Advertising cooperative phenotype through costly signals facilitates collective action" has been accepted in principle for publication in Royal Society Open Science subject to minor revision in accordance with the referee and editor suggestions. Please find their comments at the end of this email.

The reviewers and handling editors have recommended publication, but also suggest some minor revisions to your manuscript. Therefore, I invite you to respond to the comments and revise your manuscript.

Please you submit the revised version of your manuscript within 7 days (i.e. by the 29-May-2021). If you do not think you will be able to meet this date please let me know immediately.

When submitting your revised manuscript, you will be able to respond to the comments made by the referees and you should upload a file "Response to Referees". You can use this to document any changes you make to the original manuscript. In order to expedite the processing of the revised manuscript, please be as specific as possible in your response to the referees.

Full author guidelines can be found here
<https://royalsocietypublishing.org/rsos/registered-reports>.

on behalf of Professor Chris Chambers (Subject Editor, Royal Society Open Science)
openscience@royalsociety.org

Associate Editor Comments to Author (Professor Chris Chambers):

Associate Editor: 1

Comments to the Author:

The three expert reviewers who assessed the initial Stage 1 submission have returned to evaluate the revised manuscript. The good news is that all reviewers are now positive and we are within striking distance of in-principle acceptance (IPA). A final round of minor revision is necessary to address interpretation of potential results, clarification of the methodology, and potential limitations. Concerning the model equations and the comment by Reviewer 1, I think it is best to retain these in the manuscript but to, of course, ensure they are correctly specified. Provided you are able to respond comprehensively to these remaining points, IPA should be forthcoming without requiring further in-depth Stage 1 review.

Reviewer comments to Author:

Reviewer: 1

Comments to the Author(s)

The authors present a revised and much improved version of their registered report proposing an experimental investigation into cooperative tendencies, costly (ritualistic) signalling and collective action.

The biggest concern I had with the first version of this manuscript was the lack of experimental manipulation and the impossibility to detect the independent causal effect of costly signals in the assortment of cooperators. To remedy this problem the authors

now distinguish between low-cost and high-cost conditions and randomly assign participants to those conditions. So participants either choose between concealed and low-cost revealed or between concealed and high-cost revealed. The authors hypothesize that only high cost would function to assort cooperators. This manipulation should allow authors to better disentangle the effect of costly signals from other group differences and ameliorate the limitations of self-selection into experimental conditions.

The authors also clarified the theoretical rationale of their approach and included recent modeling work on costly signalling and reputation in repeated interactions (Roberts, 2020). This makes the introduction both clearer and better grounded in formal theory.

The authors have decided to include a behavioral measure of "cooperative phenotype" based on hypothetical PGG contributions instead of the self-report measure they used before. In general, I agree that behavioral measures are often more predictive of (real-world) behavior in other contexts, but I worry about the independence of this measure as an explanatory variable. It might not be surprising that individuals who indicate cooperative intentions in a hypothetical PGG will also behave more cooperatively in a real PGG. Furthermore, participants might even remember the answer given in the hypothetical scenario and might play a similar strategy in the actual game independently of their actual latent cooperative dispositions. The authors might want to comment on this potential issue.

The authors have improved their statistical notation, but there are still some issues. The link function in GLMs is not applied to the outcome but to a parameter in the likelihood. Focussing on the binomial model for H1: The outcome Y_i (0 or 1, not the probability as stated by the authors) is binomially distributed, so $Y_i \sim \text{binom}(1, p_i)$, "n=1" because there is just one "trial" per participant. We then connect the participant-specific probability p_i to the linear model as $\text{logit}(p_i) = \dots$. In my view, including the model equations is also not absolutely necessary as long as the code is accessible and the models are correctly described, but if authors decide to include equations, they should make sure they are correct.

All remaining points were addressed satisfactorily by the authors. Thank you for your effort and rigor!

Reviewer: 2

Comments to the Author(s)

The authors have considered the comments made by me and the other two reviewers; I'm satisfied with how my comments have been addressed. Considering all comments it appears that the authors have incorporated suggestions in most instances. In cases where they have not, the authors have provided a sound justification for not including changes at this time. For example, I think it's appropriate that they plan to provide possible extensions in the eventual discussion, because the proposed study is an initial attempt to establish the relationship between cooperative phenotype, costly signalling, and behaviour. I think that the three major methodological changes --- inclusion of the Fischbacher et al. (2001) behavioural assessment, replacement of an effortful condition with a low-cost condition, and more public goods game iterations --- have improved the proposed study. Some of these changes are in line with adjustment of the general framing in the introduction, which is also clear.

Two notes, neither of which prevent the manuscript being accepted. 1) Could Figure S1 be moved to the manuscript? I find it a useful reference! 2) Could it be specified in H3, Table 1 that a "future" or "follow-up" study would investigate the effect of between-group competition? This would make clear that such a design would be warranted as part of a new study if the results of the current one are not confirmed (as the authors intend) and is not necessary as part of the current study. This future work could potentially consider other explanations, as group competition might be one of several plausible contexts under which the effects of costly signals may be observed.

I look forward to seeing the results of this high quality study, which should stimulate insights and further research into the role of costly signalling in cooperation.

Reviewer: 3

Comments to the Author(s)

Thank you for inviting me to review this revised RR. My overall impression is that the manuscript has substantially improved and I am for the most part satisfied with the authors' responses to my previous comments. Below, I am describing a few remaining concerns:

1. I think that manipulating norms in a laboratory setting would be feasible, without adverse real-life consequences for participants (although mimicking real-life socialization processes would be very challenging in the lab). That said, I appreciate that the authors have re-written the introduction in a way that emphasizes cooperative phenotype and cooperative intention signaling, and de-emphasizes participation in culturally specific and extremely costly rituals. I think the current version of the introduction fits more closely with the questions that the proposed study can investigate.

2. In the revision, the authors suggest that it is not only extreme ritual behaviors that can signal a cooperative phenotype, but that more mundane behaviors, such as church attendance, can work in a similar way. This gives rise to the question of how costly a behavior needs to be in order to act as an honest signal of an underlying cooperative phenotype. I think that the authors' decision to manipulate the cost (high versus low) of signaling is a great addition to the design. But, it is possible with this design that the authors find support for their hypotheses, albeit not only for the high-cost condition. What would the authors conclude if, for example, participants with a cooperative phenotype are equally likely to signal in both the high- and low-cost conditions? Or if both low-cost and high-cost signaling are sufficient to boost contributions and stabilize cooperation? Would this go against the broader theory, or would it suggest that relatively low-cost signals might still allow assortment of cooperators (and maintenance of contributions at high levels)?

3. I think that the addition of the behavioral task to measure participants' cooperative phenotype is a significant improvement over the previous measures.

4. I have one last clarification question concerning the design: What feedback will participants receive between rounds of the PGG? In the manuscript, the authors write that "Using their remaining endowment, participants in both groups will make first iteration PGG decisions and learn about their current earnings." Does this mean that participants will not receive information about other group members' contributions? If so, this might allow selfish individuals to 'fake' a cooperative phenotype by signaling, but

keep making low contributions while remaining undetected (e.g., if they happen to be in a revealed group in which all other group members are high contributors).

Author's Response to Decision Letter for (RSOS-202202.R1)

See Appendix B.

Decision letter (RSOS-202202.R2)

Dear Mr Lang

On behalf of the Editor, I am pleased to inform you that your Manuscript RSOS-202202.R2 entitled "Advertising cooperative phenotype through costly signals facilitates collective action" has been accepted in principle for publication in Royal Society Open Science.

You may now progress to Stage 2 and complete the study as approved. Before commencing data collection we ask that you:

- 1) Update the journal office as to the anticipated completion date of your study.
- 2) Register your approved protocol on the Open Science Framework (<https://osf.io/>) or other recognised repository, either publicly or privately under embargo until submission of the Stage 2 manuscript. Please note that a time-stamped, independent registration of the protocol is mandatory under journal policy, and manuscripts that do not conform to this requirement cannot be considered at Stage 2. The protocol should be registered unchanged from its current approved state, with the time-stamp preceding implementation of the approved study design.

Following completion of your study, we invite you to resubmit your paper for peer review as a Stage 2 Registered Report. Please note that your manuscript can still be rejected for publication at Stage 2 if the Editors consider any of the following conditions to be met:

- The results were unable to test the authors' proposed hypotheses by failing to meet the approved outcome-neutral criteria.
- The authors altered the Introduction, rationale, or hypotheses, as approved in the Stage 1 submission.
- The authors failed to adhere closely to the registered experimental procedures. Please note that any deviations from the approved experimental procedures must be communicated to the editor immediately for approval, and prior to the completion of

data collection. Failure to do so can result in revocation of in-principle acceptance and rejection at Stage 2 (see complete guidelines for further information).

- Any post-hoc (unregistered) analyses were either unjustified, insufficiently caveated, or overly dominant in shaping the authors' conclusions.
- The authors' conclusions were not justified given the data obtained.

We encourage you to read the complete guidelines for authors concerning Stage 2 submissions at <https://royalsocietypublishing.org/rsos/registered-reports#ReviewerGuideRegRep>. Please especially note the requirements for data sharing, reporting the URL of the independently registered protocol, and that withdrawing your manuscript will result in publication of a Withdrawn Registration.

Once again, thank you for submitting your manuscript to Royal Society Open Science and we look forward to receiving your Stage 2 submission. If you have any questions at all, please do not hesitate to get in touch. We look forward to hearing from you shortly with the anticipated submission date for your stage two manuscript.

Kind regards,

Royal Society Open Science Editorial Office
Royal Society Open Science
openscience@royalsociety.org

on behalf of Professor Chris Chambers (Registered Reports Editor, Royal Society Open Science)
openscience@royalsociety.org

Author's Response to Decision Letter for (RSOS-202202.R2)

See Appendix C.

RSOS-202202.R3

Review form: Reviewer 1

Is the manuscript scientifically sound in its present form?

Yes

Are the interpretations and conclusions justified by the results?

Yes

Is the language acceptable?

Yes

Do you have any ethical concerns with this paper?

No

Have you any concerns about statistical analyses in this paper?

Yes

Recommendation?

Accept with minor revision

Comments to the Author(s)

In their stage-2 RR “Advertising cooperative phenotype through costly signals facilitates collective action”, the authors report the interesting results from their experimental investigation into cooperative tendencies, costly signalling and collective action.

I am happy to see that the proposed study has now been completed and, as far as I can tell, the authors closely followed their outlined procedure and analysis plan and report the results in a clear and transparent way. I am convinced that this paper will make a solid contribution to the literature and will be of interest to readers interested in ritualized behavior and the evolution of cooperation. However, before fully recommending acceptance, I have some additional comments regarding the analytical procedure and conclusions.

First, I do not fully understand why the authors use statistical categorization of individuals into different cooperative strategies instead of a more direct behavioral measure such as the (conditional) PGG contributions as predictors to test H1 and H2. The mixture model results are interesting, but using them as predictors ignores a great deal of variation that is present in the raw data. It would at least be interesting to see the results if raw contributions are used and whether the results would stay the same. Relatedly, using the classifications as predictors in regression essentially treats uncertain estimates as observed data and omits the uncertainty about cluster assignment. Ideally, the full distribution of categorization estimates would be used as predictors in the same model (or is there no uncertainty in cluster assignment?!). However, I am aware that these types of analyses are maybe not straightforward and somewhat deviate from the state-1 RR, so feel free to ignore these suggestions. In any case, the authors might want to report some more details about the finite mixture model and discuss this issue of propagating uncertainty throughout the analysis.

I also have some questions about the zero-or-one-inflated beta model. Mixture models are appropriate when there are different processes generating the data, i.e., zeros (or ones) can arise through multiple ways, not when zeros (or ones) are generated from the same process as other outcomes, even if there are many of them. Are the authors really assuming a mixture of different processes (which?) or are the results rather stemming from inter-individual differences. Do you assume, for example, that individuals who contributed either 0 or 1 did not understand the task in the same way?

Describing the mixture components of the ZOIB regression in the results section, the authors write: “3) the probability that an observation is zero given the probability that an observation is one and 4) the probability that an observation is one given the probability that an observation is zero” (page 22, lines 49-50). This is very unclear to me and I do not understand what the authors are trying to say here. Relatedly, Table 1 does not seem to be updated, as there is still the formula for the regular beta regression.

As mentioned in previous reviews, I still find the term “the cooperative phenotype” rather problematic. Human cooperation is highly flexible and context-dependent and even though stable inter-individual differences surely exist, the trait reflected in those

game choices hardly qualifies as THE cooperative phenotype, but rather corresponds to the tendency to show more cooperative behavior in the context of economic games played online with anonymous others. Additionally, the term phenotype suggests a single trait or unitary construct, whereas cooperative behavior is surely generated by a large suit of different processes. I would suggest the authors to use more careful wording that more closely corresponds to the actual behavioral measure or at least better qualify the term and/or use quotation marks.

Stylistic points:

Describing their results in the abstract, the authors only write “The results supported our predictions except for larger earnings”. I find this rather vague and non-informative and the authors may want to describe their main findings in a more substantive manner.

I am unsure about the extent to which the description and results of the pilot study belong in the main paper or whether they should better be discussed in a supplementary file. I find this section is quite lengthy and might distract from the general flow of the article.

I also find it quite hard to follow the design section. It might make it easier for readers to follow if there were some more structured subsections, etc.

I would suggest to not only refer to numbered hypotheses (H1-H4) in the results section, which forces readers to jump back and forth and makes it very difficult to keep track. It might be better to briefly describe what exactly is being tested at each point.

Finally, the title states that costly signaling of cooperative intention “facilitates collective action”. I am wondering if this is still an accurate representation of the results, since the authors did not find evidence that participants actually earned more in the revealed conditions (especially not for high costs). Maybe “Advertising cooperative phenotype through costly signals facilitates assortment of cooperators” or something along these lines might be a more accurate description of the results.

Review form: Reviewer 2

Is the manuscript scientifically sound in its present form?

Yes

Are the interpretations and conclusions justified by the results?

Yes

Is the language acceptable?

Yes

Do you have any ethical concerns with this paper?

No

Have you any concerns about statistical analyses in this paper?

No

Recommendation?

Accept with minor revision

Comments to the Author(s)

I was very much looking forward to reviewing the Stage 2 manuscript, having been involved with Stage 1. The study makes an important contribution to the literature on human cooperation and costly signalling, and will inspire experimental extensions. Generally, the manuscript is of high quality, with a solid introduction and methods, and with data that are presented adequately in both the results and the discussion. The authors are to be commended for having specified how they would adjust analyses based on the distribution of the data, as some of these analyses were required. I was able to reproduce results based on the data and code provided.

There are a few minor points, which I raise through the appraisal of the five required aspects:

1) Yes, the data are able to test the proposed hypotheses.

However, could the authors clarify the statement relating to the reduction in statistical power between the first ($n = 372$) and second ($n = 284$) phase of the study? The reduction is pointed out in the results (p22 l15), with reference to the a priori power analysis summarised in Figure S1. The authors state that 284 participants in total (i.e. 142 participants per condition) should be sufficient to detect the expected effects with 80% power. Is this assuming a moderate effect size? Figure S1 suggests 142 participants would achieve >80% power for H3 and ~80% for H4 (assuming a moderate effect size). But the actual sample size per condition is 156 (high cost) and 128 (low cost) participants --- does the lower bound of 128 participants not mean >80% power for H3 but ~75% for H4 (assuming a moderate effect size)? The reduction in sample size is not an issue: it is due to drop-outs, which are unavoidable given the approximately one week interval between phases, and the authors tried to mitigate by recruiting 138 more participants than required. However, the statement about power in the results should be made clearer and any adjustment made to the statement if necessary. Furthermore, descriptive statistics of the sample in the second stage could also be provided, to get a sense of how similar the samples are across phases. The descriptives should include SD_age and/or age range.

I was able to run and reproduce all H1--H4 analyses included in "Advertising_cooperative_V1.0.Rmd". I was unable to assess the codification of open responses about the reasons for selecting either the revealed or the concealed group, as only the original (in Czech) open responses are provided. If the authors deemed it appropriate, English translations could be provided in "Free-response_ratings.csv" to allow others to reproduce the codification. I leave it to the authors to decide whether translation would be best done either by them, through manual input in a web service, or using an R package. In line with best practice, a data dictionary should be included, describing e.g. columns B0--B20 in "Advertising_cooperative_V1.0.csv", levels in column SEX. Without a data dictionary it is not immediately clear what the variables or levels correspond to!

2) The introduction, rationale and stated hypotheses are similar to the approved Stage 1 submission. Minor edits have been made to the introduction and methods, mostly to the tense (from future to past tense). The planned analyses have not changed. The term

"strategy" is now used instead of "type" to describe individuals decisions in the first game, with the justification provided in the discussion --- I think this is appropriate.

3) The authors adhered to the registered experimental procedures.

The authors stated that the survey would collect demographic information related to education (p9, l25), but no education question appears to have been asked (based on the survey and the data). As this information was never intended to be used in confirmatory analyses, and no exploratory analyses are provided using demographic information, "education" can be removed from the design section.

4) No unregistered exploratory statistical analyses are reported. While the hypothesis table explicitly stated agnosticism about comparisons of selfish individuals with tempted individuals, could the authors include such a comparison in the SI? This could guide future studies that might consider analysing this strategy --- it appears tempted individuals are e.g. less (more) similar to selfish (cooperative) individuals in mentioning a high cost.

5) Broadly, the authors' conclusions are justified given the data, with some of the discussion focusing on how the experiment could be adjusted in future research. I would appreciate a clear summary of whether H1--H4 are supported. Some approaches in the interpretation of H3--H4 are not discussed and could be mentioned (e.g. why there was no reason to include group competition).

Minor edits:

- + introduce zero-or-one-inflated beta acronym on p11 l59
- + "give" to "given" in Figure 1 caption
- + "tree" in Table 1, H1 and H2
- + "If" to "if" p6 l17
- + "Cis" to "CIs" p17 l20 and Figure 2 caption
- + double check some of usage of "will" in the methods, e.g. "we will r" p11 l59
- + "contra-productive" to "counterproductive" p25 l14

Review form: Reviewer 3

Is the manuscript scientifically sound in its present form?

Yes

Are the interpretations and conclusions justified by the results?

No

Is the language acceptable?

Yes

Do you have any ethical concerns with this paper?

No

Have you any concerns about statistical analyses in this paper?

No

Recommendation?

Major revision

Comments to the Author(s)

Thank you for inviting me to review this Stage-2 registered report.

My overall impression is that the Stage-2 manuscript is clearly written, that the authors followed the pre-registered plan, and that the study resulted in interesting findings. I have one concern regarding the reporting and interpretation of findings on which I elaborate below. Based on this concern, I also ask the authors to provide some additional details on the results.

As I had mentioned in a previous revision round, I thought that the manipulation of costs of signaling was a great addition to the study, but one that left open two possibilities. The first was that the authors would find support for their hypotheses specifically in the high-cost condition. In my understanding, this was indeed their expectation, e.g., see current discussion (emphasis is mine) "...we investigated whether participants with the cooperative phenotype would choose to reveal the quality of their phenotype using a *costly* signal and whether this signal would be associated with contributions to a group cooperative effort. Furthermore, we expected that this relationship would help *only* in the case of a costly signal." A second possibility was that the hypothesized patterns would be observed both for high- and low-cost signals. My interpretation of the reported findings is that they support this second possibility.

Specifically, going back to the pre-registered analysis plan, my impression is that hypotheses 1-3 referred to interaction effects. However:

- When testing hypothesis 1, the authors do not find an interaction effect of low-cost \times selfish strategy (the CI includes 0). Strictly speaking, this result does not provide support for hypothesis 1 – i.e., the effect of cooperative phenotype on selecting the revealed group does not depend on whether the cost is high or low. In my opinion, this should be more explicitly noted. Nevertheless, the authors go on to interpret the simple effect of strategy in the high-cost condition, reporting that participants with a selfish strategy were less likely to chose the revealed group than the concealed group in the high-cost condition. Then, for the sake of completeness, I would also like to see the simple effect of selfish strategy in the low-cost condition.
- When testing hypothesis 2, the authors find support for the hypothesized interaction between low-cost \times selfish on mentioning the costs of revealing. As before, they provide information on the simple effect of strategy in the high-cost condition. It would be helpful to also know what the simple effect of strategy was in the low-cost condition.
- When testing hypothesis 3, in two out of the three reported analyses, the interaction between low cost \times concealed predicting contributions is not statistically significant (i.e., in the models looking at percent of contributions and the probability of sending everything). Again, my impression is that these results do not provide support for hypothesis 3 (as it was pre-registered) – i.e., the effects of being in the revealed group on contributions do not depend on whether the cost is high or low. This could be more explicitly noted. Finally, the authors provide results for the effects of being in the revealed

group on contributions in the high-cost condition, but not in the low-cost condition. For the sake of completeness, and to better understand the pattern of results in the two high- and low- cost conditions, it would be helpful to see simple effects in each of these conditions. The same applies to hypothesis 4.

In sum, I would like to see a more explicit mention of whether the pattern of results obtained in each analysis matches (or not) the hypotheses as pre-registered. It seems to me that, in many cases, the results do not provide support for the hypothesized interaction effects. Instead, there is support for simple effects consistent with the authors' ideas in the high-cost condition. Presenting the simple effects also in the low-cost condition would give readers a clearer idea about the full pattern of findings.

Decision letter (RSOS-202202.R3)

Dear Mr Lang:

On behalf of the Editor, I am pleased to inform you that your Stage 2 Registered Report RSOS-202202.R3 entitled "Advertising cooperative phenotype through costly signals facilitates collective action" has been deemed suitable for publication in Royal Society Open Science subject to minor revision in accordance with the referee suggestions. Please find the referees' comments at the end of this email.

The reviewers and Subject Editor have recommended publication, but also suggest some minor revisions to your manuscript. We invite you to respond to the comments and revise your manuscript. Below the referees' and Editors' comments (where applicable) we provide additional requirements. Final acceptance of your manuscript is dependent on these requirements being met. We provide guidance below to help you prepare your revision.

Please submit your revised manuscript and required files (see below) no later than 30 days from today's (ie 01-Apr-2022) date. Note: the ScholarOne system will 'lock' if submission of the revision is attempted 30 or more days after the deadline. If you do not think you will be able to meet this deadline please contact the editorial office immediately.

on behalf of Professor Chris Chambers
 (Registered Reports Editor, Royal Society Open Science)
 openscience@royalsociety.org

Associate Editor Comments to Author (Professor Chris Chambers):

The three reviewers who assessed the manuscript at Stage 1 kindly returned to evaluate the Stage 2 submission. As you will see, all are detailed and helpful, and are broadly positive about the final manuscript, while also noting some points that would benefit from further consideration and revision; these include ensuring that the discussion makes clear the level of support for the specific hypotheses, some suggestions for potential exploratory analyses (which are optional), and clarification of methodological details. In revising please do not make any changes to the introduction and methods unless doing so is necessary to correct factual errors or avoiding misunderstandings. Consideration of methodological limitations can be included in the Discussion.

Comments to Author:

Reviewer: 1

Comments to the Author(s)

In their stage-2 RR "Advertising cooperative phenotype through costly signals facilitates collective action", the authors report the interesting results from their experimental investigation into cooperative tendencies, costly signalling and collective action.

I am happy to see that the proposed study has now been completed and, as far as I can tell, the authors closely followed their outlined procedure and analysis plan and report the results in a clear and transparent way. I am convinced that this paper will make a solid contribution to the literature and will be of interest to readers interested in ritualized behavior and the evolution of cooperation. However, before fully recommending acceptance, I have some additional comments regarding the analytical procedure and conclusions.

First, I do not fully understand why the authors use statistical categorization of individuals into different cooperative strategies instead of a more direct behavioral measure such as the (conditional) PGG contributions as predictors to test H1 and H2. The mixture model results are interesting, but using them as predictors ignores a great deal of variation that is present in the raw data. It would at least be interesting to see the results if raw contributions are used and whether the results would stay the same. Relatedly, using the classifications as predictors in regression essentially treats uncertain estimates as observed data and omits the uncertainty about cluster assignment. Ideally, the full distribution of categorization estimates would be used as predictors in the same model (or is there no uncertainty in cluster assignment?!). However, I am aware that these types of analyses are maybe not straightforward and somewhat deviate from the state-1 RR, so feel free to ignore these suggestions. In any case, the authors might want to report some

more details about the finite mixture model and discuss this issue of propagating uncertainty throughout the analysis.

I also have some questions about the zero-or-one-inflated beta model. Mixture models are appropriate when there are different processes generating the data, i.e., zeros (or ones) can arise through multiple ways, not when zeros (or ones) are generated from the same process as other outcomes, even if there are many of them. Are the authors really assuming a mixture of different processes (which?) or are the results rather stemming from inter-individual differences. Do you assume, for example, that individuals who contributed either 0 or 1 did not understand the task in the same way?

Describing the mixture components of the ZOIB regression in the results section, the authors write: “3) the probability that an observation is zero given the probability that an observation is one and 4) the probability that an observation is one given the probability that an observation is zero” (page 22, lines 49-50). This is very unclear to me and I do not understand what the authors are trying to say here. Relatedly, Table 1 does not seem to be updated, as there is still the formula for the regular beta regression.

As mentioned in previous reviews, I still find the term “the cooperative phenotype” rather problematic. Human cooperation is highly flexible and context-dependent and even though stable inter-individual differences surely exist, the trait reflected in those game choices hardly qualifies as THE cooperative phenotype, but rather corresponds to the tendency to show more cooperative behavior in the context of economic games played online with anonymous others. Additionally, the term phenotype suggests a single trait or unitary construct, whereas cooperative behavior is surely generated by a large suit of different processes. I would suggest the authors to use more careful wording that more closely corresponds to the actual behavioral measure or at least better qualify the term and/or use quotation marks.

Stylistic points:

Describing their results in the abstract, the authors only write “The results supported our predictions except for larger earnings”. I find this rather vague and non-informative and the authors may want to describe their main findings in a more substantive manner.

I am unsure about the extent to which the description and results of the pilot study belong in the main paper or whether they should better be discussed in a supplementary file. I find this section is quite lengthy and might distract from the general flow of the article.

I also find it quite hard to follow the design section. It might make it easier for readers to follow if there were some more structured subsections, etc.

I would suggest to not only refer to numbered hypotheses (H1-H4) in the results section, which forces readers to jump back and forth and makes it very difficult to keep track. It might be better to briefly describe what exactly is being tested at each point.

Finally, the title states that costly signaling of cooperative intention “facilitates collective action”. I am wondering if this is still an accurate representation of the results, since the authors did not find evidence that participants actually earned more in the revealed conditions (especially not for high costs). Maybe “Advertising cooperative phenotype through costly signals facilitates assortment of cooperators” or something along these lines might be a more accurate description of the results.

Reviewer: 2

Comments to the Author(s)

I was very much looking forward to reviewing the Stage 2 manuscript, having been involved with Stage 1. The study makes an important contribution to the literature on human cooperation and costly signalling, and will inspire experimental extensions. Generally, the manuscript is of high quality, with a solid introduction and methods, and with data that are presented adequately in both the results and the discussion. The authors are to be commended for having specified how they would adjust analyses based on the distribution of the data, as some of these analyses were required. I was able to reproduce results based on the data and code provided.

There are a few minor points, which I raise through the appraisal of the five required aspects:

1) Yes, the data are able to test the proposed hypotheses.

However, could the authors clarify the statement relating to the reduction in statistical power between the first ($n = 372$) and second ($n = 284$) phase of the study? The reduction is pointed out in the results (p22 l15), with reference to the a priori power analysis summarised in Figure S1. The authors state that 284 participants in total (i.e. 142 participants per condition) should be sufficient to detect the expected effects with 80% power. Is this assuming a moderate effect size? Figure S1 suggests 142 participants would achieve >80% power for H3 and ~80% for H4 (assuming a moderate effect size). But the actual sample size per condition is 156 (high cost) and 128 (low cost) participants --- does the lower bound of 128 participants not mean >80% power for H3 but ~75% for H4 (assuming a moderate effect size)? The reduction in sample size is not an issue: it is due to drop-outs, which are unavoidable given the approximately one week interval between phases, and the authors tried to mitigate by recruiting 138 more participants than required. However, the statement about power in the results should be made clearer and any adjustment made to the statement if necessary. Furthermore, descriptive statistics of the sample in the second stage could also be provided, to get a sense of how similar the samples are across phases. The descriptives should include SD_age and/or age range.

I was able to run and reproduce all H1--H4 analyses included in "Advertising_cooperative_V1.0.Rmd". I was unable to assess the codification of open responses about the reasons for selecting either the revealed or the concealed group, as only the original (in Czech) open responses are provided. If the authors deemed it appropriate, English translations could be provided in "Free-response_ratings.csv" to allow others to reproduce the codification. I leave it to the authors to decide whether translation would be best done either by them, through manual input in a web service, or using an R package. In line with best practice, a data dictionary should be included, describing e.g. columns B0--B20 in "Advertising_cooperative_V1.0.csv", levels in column SEX. Without a data dictionary it is not immediately clear what the variables or levels correspond to!

2) The introduction, rationale and stated hypotheses are similar to the approved Stage 1 submission. Minor edits have been made to the introduction and methods, mostly to the tense (from future to past tense). The planned analyses have not changed. The term "strategy" is now used instead of "type" to describe individuals decisions in the first game, with the justification provided in the discussion --- I think this is appropriate.

3) The authors adhered to the registered experimental procedures.

The authors stated that the survey would collect demographic information related to education (p9, l25), but no education question appears to have been asked (based on the survey and the data). As this information was never intended to be used in confirmatory analyses, and no exploratory analyses are provided using demographic information, "education" can be removed from the design section.

4) No unregistered exploratory statistical analyses are reported. While the hypothesis table explicitly stated agnosticism about comparisons of selfish individuals with tempted individuals, could the authors include such a comparison in the SI? This could guide future studies that might consider analysing this strategy --- it appears tempted individuals are e.g. less (more) similar to selfish (cooperative) individuals in mentioning a high cost.

5) Broadly, the authors' conclusions are justified given the data, with some of the discussion focusing on how the experiment could be adjusted in future research. I would appreciate a clear summary of whether H1--H4 are supported. Some approaches in the interpretation of H3--H4 are not discussed and could be mentioned (e.g. why there was no reason to include group competition).

Minor edits:

- + introduce zero-or-one-inflated beta acronym on p11 l59
- + "give" to "given" in Figure 1 caption
- + "tree" in Table 1, H1 and H2
- + "If" to "if" p6 l17
- + "Cis" to "CIs" p17 l20 and Figure 2 caption
- + double check some of usage of "will" in the methods, e.g. "we will r" p11 l59
- + "contra-productive" to "counterproductive" p25 l14

Reviewer: 3

Comments to the Author(s)

Thank you for inviting me to review this Stage-2 registered report.

My overall impression is that the Stage-2 manuscript is clearly written, that the authors followed the pre-registered plan, and that the study resulted in interesting findings. I have one concern regarding the reporting and interpretation of findings on which I elaborate below. Based on this concern, I also ask the authors to provide some additional details on the results.

As I had mentioned in a previous revision round, I thought that the manipulation of costs of signaling was a great addition to the study, but one that left open two possibilities. The first was that the authors would find support for their hypotheses specifically in the high-cost condition. In my understanding, this was indeed their expectation, e.g., see current discussion (emphasis is mine) "...we investigated whether participants with the cooperative phenotype would choose to reveal the quality of their phenotype using a *costly* signal and whether this signal would be associated with contributions to a group cooperative effort. Furthermore, we expected that this relationship would help *only* in the case of a costly signal.*" A second possibility was that the hypothesized patterns

would be observed both for high- and low-cost signals. My interpretation of the reported findings is that they support this second possibility.

Specifically, going back to the pre-registered analysis plan, my impression is that hypotheses 1-3 referred to interaction effects. However:

- When testing hypothesis 1, the authors do not find an interaction effect of low-cost × selfish strategy (the CI includes 0). Strictly speaking, this result does not provide support for hypothesis 1 – i.e., the effect of cooperative phenotype on selecting the revealed group does not depend on whether the cost is high or low. In my opinion, this should be more explicitly noted. Nevertheless, the authors go on to interpret the simple effect of strategy in the high-cost condition, reporting that participants with a selfish strategy were less likely to chose the revealed group than the concealed group in the high-cost condition. Then, for the sake of completeness, I would also like to see the simple effect of selfish strategy in the low-cost condition.

- When testing hypothesis 2, the authors find support for the hypothesized interaction between low-cost × selfish on mentioning the costs of revealing. As before, they provide information on the simple effect of strategy in the high-cost condition. It would be helpful to also know what the simple effect of strategy was in the low-cost condition.

- When testing hypothesis 3, in two out of the three reported analyses, the interaction between low cost × concealed predicting contributions is not statistically significant (i.e., in the models looking at percent of contributions and the probability of sending everything). Again, my impression is that these results do not provide support for hypothesis 3 (as it was pre-registered) – i.e., the effects of being in the revealed group on contributions do not depend on whether the cost is high or low. This could be more explicitly noted. Finally, the authors provide results for the effects of being in the revealed group on contributions in the high-cost condition, but not in the low-cost condition. For the sake of completeness, and to better understand the pattern of results in the two high- and low- cost conditions, it would be helpful to see simple effects in each of these conditions. The same applies to hypothesis 4.

In sum, I would like to see a more explicit mention of whether the pattern of results obtained in each analysis matches (or not) the hypotheses as pre-registered. It seems to me that, in many cases, the results do not provide support for the hypothesized interaction effects. Instead, there is support for simple effects consistent with the authors' ideas in the high-cost condition. Presenting the simple effects also in the low-cost condition would give readers a clearer idea about the full pattern of findings.

===PREPARING YOUR MANUSCRIPT===

one version should clearly identify all the changes that have been made (for instance, in coloured highlight, in bold text, or tracked changes);

===PREPARING YOUR REVISION IN SCHOLARONE===

- An editable file of all figure and table captions.
- Note: you may upload the figure, table, and caption files in a single Zip folder.
- Any electronic supplementary material (ESM).
 - If you are requesting a discretionary waiver for the article processing charge, the waiver form must be included at this step.
 - If you are providing image files for potential cover images, please upload these at this step, and inform the editorial office you have done so. You must hold the copyright to any image provided.
 - A copy of your point-by-point response to referees and Editors. This will expedite the preparation of your proof.

- Ensure that your data access statement meets the requirements at <https://royalsociety.org/journals/authors/author-guidelines/#data>. You should ensure that you cite the dataset in your reference list. If you have deposited data etc in the Dryad repository, please only include the 'For publication' link at this stage. You should remove the 'For review' link.
- If you are requesting an article processing charge waiver, you must select the relevant waiver option (if requesting a discretionary waiver, the form should have been uploaded, see 'File upload' above).
- If you have uploaded any electronic supplementary (ESM) files, please ensure you follow the guidance at <https://royalsociety.org/journals/authors/author-guidelines/#supplementary-material> to include a suitable title and informative caption. An example of appropriate titling and captioning may be found at https://figshare.com/articles/Table_S2_from_Is_there_a_trade-off_between_peak_performance_and_performance_breadth_across_temperatures_for_aerobic_scope_in_teleost_fishes_/3843624.

Author's Response to Decision Letter for (RSOS-202202.R3)

See Appendix D.

Decision letter (RSOS-202202.R4)

Dear Martin,

It is a pleasure to accept your manuscript entitled "Advertising cooperative phenotype through costly signals facilitates collective action" in its current form for publication in Royal Society Open Science.

Thank you for your fine contribution. On behalf of the Editors of Royal Society Open Science, we look forward to your continued contributions to the journal.

on behalf of Professor Chris Chambers (Subject Editor)
openscience@royalsociety.org

Appendix A

Dear Prof. Chambers,

we are grateful for the chance to revise our manuscript, which we believe is now much improved due to the reviewers' excellent suggestions. Below, we grouped reviewers' concerns and suggestions into logical chunks and, wherever feasible, addressed them together.

1. Concerns related to theory

R1: I have questions about the application of costly signaling theory to religious rituals in general. The peacock's tail only truthfully signals genetic quality because there is a shared genetic mechanism producing the signal and other (hidden) qualities. I do not see how participation in rituals is connected to other collective endeavors in such a direct way (as the authors also mention on p. 4, lines 57-59). Wouldn't individuals that only participate in collective ritual and then free-ride in actual community activities always end up with the highest payoff? Why would only truly cooperative individuals be able to participate in costly rituals? The authors mention Sosis' (2003) explanation in terms of perceived costs, but it is unclear (at least to me) how such psychological costs might affect real evolutionary dynamics that operate on material payoffs to different strategies. Recent theory suggests that costly helping behavior can evolve into an honest signal when it is adaptive for cooperators to invest in the long-term benefits of a reputation for helping (Roberts, 2020, Honest signaling of cooperative intentions). Such reputation effects in repeated interactions might be a more plausible mechanism than perceived psychological costs.

Answer: As R1 notes, we agree that applying the costly signaling theory to human behavior, especially to their mental commitment to cooperative interactions, is not straightforward. Indeed, the assumed hidden quality is not phenotypically linked to the signaled trait such that the low-quality signalers cannot perform the high-quality signal (as is the case of, e.g., ungulate stotting), nor it is linked to differential costs of the production of the high-quality signals (as is the case of, e.g., peacock's train). The issue here is that the signalled quality is of psychological rather than strictly physiological nature and, thus, the link to the physiological/material costs of a signal must be only indirect. It also means that the hidden psychological quality is subject to ontogenetic processes, is more flexible than morphological phenotypes, and must be maintained during the lifetime.

Our design aimed to capture the more stable aspect of this quality and called it a 'cooperative phenotype'. We expected that people high on cooperative predisposition should be more likely to follow general cooperative norms and be willing to pay a cost to increase the chances of assortment with like-minded individuals. The cost itself should be seen as less costly by cooperative participants because they would see the overall cooperative benefit after several rounds of PGG with like-minded co-players. In this prediction, we recognized that the decision to signal is crucially dependent on the iterated nature of PGG because one-shot PGG (or any other economic interaction) might be invaded by free-riders who would still perceive paying the initial cost as worth the interaction with cooperators. We are grateful to R1 for pointing us to the Roberts (2020) paper because it neatly captures and formalizes this idea. We still maintain that there is an underlying hidden quality that people signal, but the repeated interactions are crucial for the signal to work.

Embracing Roberts' theory does not mean that we think that the model presented by Sosis (2003) is wrong, but we now see that the model does not fit our design as good as the model presented by Roberts. Sosis' model aims to capture the fact that the cooperative phenotype is manifested through adherence to norms of a particular cultural group (which can change during the lifetime) and addresses the more flexible part of the psychological commitment to a particular group, which needs to be developed over time via internalization. Sosis argues that by regularly participating in communal rituals, participants signal their norm commitment to others and, crucially, also to oneself; and it is partially through this self-communicating function that the hidden quality is maintained.

It might be possible for free-riders to once perform the ritual without adhering to the values the ritual indicates and, in turn, exploit the ritual community. However, the fact that rituals are performed regularly should lead the free-rider either to discontinue the ritual practice that is dissonant with their lack of commitment or to change the underlying quality (i.e., commit to group norms and cooperate). Cost perception is subject to long-term internalization processes related to a gradual modification of commitment to norms of a specific group (either by being born to that group or by conversion). However, we agree that our design cannot speak to this long-term internalization process and change in perception.

Action: We substantially re-wrote the introduction section to make a case for a general model of cooperative costly signaling in iterated scenarios, discussing the role of intention signaling as described by Roberts (2020). In doing so, we reduced the part of the introduction section focused on Sosis' model and will present the model in more detail in the discussion section as a possible extension of our design that could be taken up by other researchers focused on the long-term commitment to particular groups. Importantly, by incorporating the behavioral assessment of cooperative phenotype (as in Fischbacher, Gächter, & Fehr, 2001; see below), we will be able to test whether participants with cooperative phenotype would choose to signal their phenotype (consonant with the costly signaling model) or whether the signal would be used only as a signal of intention independent of the underlying cooperative phenotype (as suggested by Roberts' paper). We also added a hypothesis that should facilitate differentiating between these competing scenarios.

R3: I think the proposed paradigm might not be ideal to answer this question... because the behaviors of interest are culturally prescribed and, as the authors note, are based on long-term socialization processes and internalized norms, whereas signaling behaviors in the proposed studies lack cultural meaning and do not appear ritualized.

Answer: Please see our previous answer to R1. Norm internalization would be extremely difficult if not unethical to conduct in a laboratory setting. That is, manipulating the internalization of cooperative norms in participants could have severe real-life consequences. Therefore, we opted to focus on an already existing cooperative phenotype. While we do not manipulate the quality of the phenotype, we harness the existing variability in the expression of this phenotype to test whether the quality interacts with costly signals in stabilizing cooperative behavior in anonymous individuals. In this respect, we circumvent the need to manipulate norm internalization while still testing the relationship between hidden quality and costly signaling.

Action: We shifted the focus in the introduction section from norm internalization proposed by Sosis (2003) to the concept of cooperative phenotype to show that the variation in cooperative phenotype should co-vary with the willingness to reveal this phenotype. Our design now better reflects the theoretical introduction by focusing on the general cooperative intentions in an iterated PGG (as suggested by R1).

R2: The statement “by including reputation and experience as decision-making mechanisms” (line 28-29, pg. 7) requires further discussion. How are these mechanisms present in the proposed study?

Answer: We are grateful to R2 for bringing our attention to this imprecise expression. We meant that the repeated nature of our design would stimulate these mechanisms. Participants would decide in the rounds following the first round based on their experience of contributions from other players in previous rounds and care about their reputation for future rounds.

Action: We clarified this sentence to reflect our design better. The sentence now reads: “...iterated PGG that better reflects the dynamics of real-world collective action where cooperative interactions between specific individuals are often repeated”.

R2: Mention strategic concerns or signs of misunderstanding present in PGGs, as discussed in e.g. Burton-Chellew, El Mouden & West (2017) <https://doi.org/10.1098/rspb.2017.0689>. Strategic concerns might be particularly pertinent as “ritual sacrifice” (as per the proposed study) could be considered not as a signal of hidden quality but as a method to induce cooperation in others. Additionally, these motivations could be assessed in the written rationale provided by participants.

Answer: To test whether the signal relates to a hidden quality or is just a method to induce cooperation, we added a behavioral assessment of cooperative phenotype using the method suggested by Fischbacher, Gächter, and Fehr (2001) in addition to the written rationale for choosing a group. We expect that in the PGG scenario proposed by Fischbacher et al., participants high on the cooperative phenotype would cooperate in the cooperative scenarios and then choose to signal their quality. However, if participants in this PGG scenario would not cooperate and then choose to signal and cooperate in the iterated PGG, this could indeed be a method to induce cooperation in others.

Action: We now implement the Fischbacher et al. behavioral method to assess participants’ cooperative phenotypes before the main PGG. This method will allow us to test whether there is a connection between a hidden quality, willingness to signal, and cooperation in PGG. We also addressed this concern in the introduction section, cited the relevant paper suggested by R2, and added an interpretation to H1 results that reflect these competing explanations.

R2: Not all costly signals involve large costs and/or extreme behaviour, often taking more subtle forms. Furthermore, signals might be directed at particular individuals. Could the authors comment on these aspects in the introduction? The former seems particularly relevant given that Study 2 involves a time consuming task, which could be considered a more discrete signal relative to sacrificing money.

Perhaps refer readers to Bird, Ready & Power (2018) <https://doi.org/10.1038/s41562-018-0298-3>.

R3: I think the proposed paradigm might not be ideal to answer this question....because the costs involved in the aforementioned ritualized behaviors seem extremely high (and this is exactly what makes these behaviors so puzzling), whereas the costs paid by participants to signal in the proposed studies are very low by comparison;

Answer: We grouped the comments of R2 and R3 here because they both relate to the size of the cost to which participants will be subjected. We agree with R2 that not all signals need to be extremely costly and that such signals would be rather exceptional. While the extreme rituals are puzzling, they are just among the many puzzling behaviors related to ritual context. Therefore, even less extreme signals (such as monetary sacrifice in our study) should produce the expected effect. Moreover, it is important to note that economic games are an imprecise simulation of the complex social reality, and the costs/earnings are set up to resemble real-life situations but not to replicate them.

Action: We changed the framing of our introduction to de-emphasize the examples of extremely costly signals and explain why even less spectacular rituals (e.g., weekly mass attendance) might be a cost that individuals pay to signal their hidden quality. We also explain that some signals may be relatively subtle and cite the suggested paper. Also, note that we do not plan to conduct the second study (see below).

2. Concerns related to design

R1: The authors set out to test an elaborate mechanism linking cooperative phenotypes, costly rituals and collective action. I do not see how the proposed design is appropriate to test any causal relationships due to a lack of experimental manipulation (or other identification strategy). In particular, the title states that it is the advertising of cooperative phenotypes through costly rituals that facilitates collective action. However, this design is unable to isolate the causal effect of costly signaling and disentangle it from any other differences between experimental conditions and pre-existing differences. It could very well be, for instance, that more cooperative individuals (as measured through self-report) behave more cooperatively and, thus, earn more in the experiment without any additional effect of costly ritual signals. To get at the causal effects, the design would require more experimental conditions and true experimental variation of costly signals (maybe also of varying strengths).

Answer: Indeed, we believe that more cooperative individuals would behave more cooperatively and earn more in the experiment. The crucial insight is that cooperators would not earn more if they could not be sure about the cooperative intentions of other players. It is in these situations when the signal becomes crucial. The signal is not assumed to make people more cooperative (this is why participants self-select into groups rather than being assigned to groups). However, we agree that our design does not utilize manipulation to directly test the effect of the signal cost on cooperative assortment. It may be that more cooperative individuals would select the revealed group for reasons unrelated to their underlying cooperative phenotype (although the analysis of reasons for choosing the revealed group in our pilot data shows otherwise). To control for this possible confound, we now include two conditions

with a different cost: low and high, assuming that only the high-cost condition would assort cooperators and lead to their larger earnings.

Action: We changed our design to include experimental manipulation of cost. Specifically, we will include two “revealed” conditions - low-cost and high-cost to experimentally assess the causal role of costly signals in assorting cooperators. While all participants will choose between “concealed” and “revealed” groups, half of our participants will be informed that the “revealed” group mandates to sacrifice 15% of their endowment before each PGG iteration, and the other half will be informed that the mandated sacrifice equals 2.5% of their endowment before each PGG iteration. Note that we also increased the planned sample size to reflect these additional comparisons to 320 participants. As a consequence, we had to put aside for the moment testing the costly signaling hypothesis with a non-monetary medium in Study 2 because the sample size for Study 2 would also need to be increased, and we do not have sufficient funding to conduct two studies with such a large sample. We hope to conduct this study in the future, however.

R1: The authors call their main explanatory variable “cooperative phenotype”. In contrast to the original paper (Peysakhovich Nowak & Rand, 2014), which also includes behavioral measures from multiple games, authors here rely only on self-reported measures of cooperative attitudes which might themselves be partly influenced by social desirability and value signaling. An assortment of cooperators into the signaling condition might simply arise because some individuals have a greater desire to present themselves as cooperative or “good” people, not because actual cooperators use costly signals to assort. In any case, the authors might want to use a different name to refer to self-reported cooperative values which, at best, correlate with real cooperative behavior.

R2: Can the authors confirm that the trustworthiness scale will not be included in the proposed analysis? If so, the authors should justify why the trustworthiness scale is being included in the proposed study.

R3: Further, if they decide to use this measure in their main studies, they should report all results when using trustworthiness as an outcome measure (in addition to results with the cooperative values scale as the outcome measure).

The first concerns the measures used to assess the ‘cooperative phenotype.’ While I see that these measures have been shown to correlate with cooperative behavior in experimental games, I am unsure about the extent to which they have been properly validated. As the authors note based on their pilot data, not all items of the cooperative values scale seem to tap a common underlying factor; further, some of the items on that scale could be measuring participants’ political orientation, besides their general cooperative tendencies. Alternative to the cooperative values and trustworthiness questionnaires, the authors could consider using well-validated measures of Social Value Orientations (e.g., the SVO Slider scale; Murphy et al., 2011). This would have the added benefit of allowing them to distinguish between joint-gain maximizers, altruists, and individuals who are inequality averse in their sample. Such distinctions may be relevant to the interpretation of results as, at least in the pilot, some high-contributing participants seem to choose the ‘concealed’ group to avoid wasting resources that can otherwise benefit the collective.

Answer: First, while we generally agree with R1 that self-reported measures do not necessarily reflect actual behavior they refer to, we chose the measure of cooperative phenotype exactly because it correlates well with behavior as shown by Peysakhovich et al. Of course, we do not know whether the

self-reported measure would show the same correlation in our population. The case could be made that our results are less precise because the Czech student population exhibits more desirability bias than the original population studied by Peysakhovich et al. Furthermore, as R3 notes, the scale's reliability is not overwhelming. To remedy these concerns, we implemented into our procedure a behavioral measure of cooperative phenotype suggested by Fischbacher, Gächter, and Fehr (2001). This measure will allow us to discern different types of cooperative phenotypes while avoiding the possible biases inherent in self-reports. Since we believe that this behavioral measure is superior to the survey measures used in the pilot study, we will not collect the self-reported data.

Action: Rather than asking participants about their cooperative behavior, we now ask them to select a hypothetical contribution to the common pool in various scenarios, from which one will be paid (hence, over-reporting cooperative intentions would be costly for free-riders). Furthermore, in the discussion section of the main text, we will suggest using the SVO slider scale in future research to get a more nuanced view of participants' motivations.

R1: The previous concern is exacerbated by the rather explicit group description which states that participants will (or will not) sacrifice part of their endowment “to demonstrate intentions regarding the size of contribution to the common pool”. In order to test the prediction that costly rituals function (in an evolutionary sense) to assort cooperators, the adaptive logic should emerge endogenously from behavior and participants should not be told explicitly what the function of the costly signal might be.

Answer: We agree with R1 that the canonical costly signaling model does not need to assume explicit knowledge about the role of the signal. Instead, the covariance between the hidden quality (manifested in later reproductive success) and the revealed phenotype (costly signal) is recognized by receivers of the signal over (extended) evolutionary time. However, in this case, the application of costly signaling theory to humans again diverges because cultural groups often make this link explicit or at least routed through a more general symbol that encapsulates the norms. Thus, participating in a painful ritual dedicated to gods indicates a belief in the deity and, by extension, commitment to abide by the norms that the deity mandates. Some rites of passage are explicitly associated with new duties that the initiates need to follow. Taking part in a Catholic mass is associated with following the Ten Commandments that explicitly stipulate the norms. There are many other examples like this.

Apart from these ethnographic observations, we were also worried that leaving the cost in one group without any additional comment might be incomprehensible for the participants. Thus, we decided to associate the cost and norm explicitly.

Action: We retained the explicit association between cost and norms in group descriptions. However, in the discussion section, we will discuss further extensions of our design that could test whether such an association may arise naturally over ‘generations’ of players.

R2: The four hypotheses in Table 1 are clear and specific (although see comment 6). Because they relate to the experimental design they are more accurate than the initial verbal description of the hypotheses. I think the terms “ritual sacrifice” and “group norms” in the verbal description can be linked more explicitly to the experiment. My questions surrounding these two terms are:

a) ritual sacrifice: it is clear that the one-time decision to discard part of the endowment represents a sacrifice. Is the ritual element represented by the sacrifice occurring before each of the three rounds?
b) group norms: it is unclear how choosing to cooperate in the revealed group represents adherence to cooperative norms. The revealed and the concealed group both include the instruction “it is expected that every member should contribute as much as possible”. Thus there is no priming difference. The only difference in expectations would come from participants self-selecting into groups, with the expectation that others will e.g. cooperate in the revealed group. Even so, there is no established norm as such, at least not in the first round of the PGG. Could the authors justify the use of the term “norms”?

Answer: Regarding sacrifice (question “a”), yes, the sacrifice of money occurs before every round of PGG.

Action: We now highlight this fact in several places in the manuscript.

Answer: Regarding group norms (question “b”), we believe that they are specified by the quote R2 highlighted in their comment (“it is expected that every member should contribute as much as possible”). This is the rule that group members should abide by, and the other reviewers seem to agree with us that this is a sufficient formulation to act as a group norm. In this sense, cooperation in both the revealed and concealed groups would represent adherence to cooperative norms. However, we aimed to avoid using the word “norm” or “rule” as not to prime participants with normative behavior.

Action: We now identify the instruction to contribute as much as possible as a group norm in the manuscript to clarify why we believe that the artificially created groups have norms.

R3: Another related concern is that (c) many ritual performances seem to signal norm abidance rather than cooperative intentions per se, but in this study the only ‘norm’ participants are exposed to (and can signal abidance to) is a norm of cooperation. (A more stringent test of the idea that ritualized behaviors per se signal an underlying cooperative phenotype, would be to examine the effects of costly following of norms that are not explicitly related to cooperation. Would individuals that signal abidance to any group norm contribute more to collective action?)

Answer: This is an intriguing suggestion and would constitute an important follow-up to discern whether participants signal their adherence to norms (cooperative or otherwise) or whether they just signal their desire to engage in cooperative interactions. We believe that if the cooperative phenotype is associated with norm adherence, this adherence should be manifested by normative behavior in various domains, possibly even in domains that are maladaptive. This conclusion can be extrapolated from related research on the role of punishment in norm stabilization, which shows that punishment can stabilize both adaptive and maladaptive norms (although only the former would survive in the long run; see Richerson & Henrich, 2012).

Action: If our manuscript proceeds to Stage-2 registered report, we would describe this extension of our research in the discussion section. For now, we opt to first empirically establish the relationship between cooperative phenotype, costly signaling, and cooperation; and only later investigate what exactly is being signaled (cooperative intentions vs norm adherence).

R2: Include a statement anticipating how the results might be affected by the sample of undergraduate students from a particular European country. Furthermore, does the population either perform costly signals or participate in religious or other committed groups in ways that might affect the results, or in ways that translate to other contexts? Similarly, the abstract should mention the nationality of the sample.

Would it be worth asking individuals about religion and religious participation? These measures could be additional covariates included in the second step for each analysis. Alternatively, these covariates could be included as descriptive sample characteristics, to get a sense of whether or not this sample engages in rituals that the experiment is meant to represent. I leave it to the authors to decide whether the inclusion of such variables is warranted!

Answer: We agree that the specificity of our sample should be explicitly mentioned. The Czech population displays a relatively low level of religiosity (or, at least, a low level of affiliation to religious organizations), so engaging in costly religious activities would likely be rare in our sample and not a confounding factor. Moreover, we believe that while some of the theoretical background for our current study comes from the study of religious rituals, our proposition is broad enough that it should not be limited to (and confounded by) religious participants. As a matter of fact, testing our hypotheses on a population largely unfamiliar with costly religious signals should present a more robust test of our hypothesis.

Action: We now describe our sample in the abstract and provide more details on the general characteristics of the Czech student population relevant for the current study in the Methods section.

R3: I see how including more rounds can allow a better understanding of cooperation dynamics in the 'revealed' and 'concealed' groups. It is reasonable to expect that cooperation can be better maintained in the 'revealed' group, but that it will break down over time in the 'concealed' group. However, I am not sure that three rounds are enough to observe this pattern and would suggest that the authors consider studying interactions over a larger number of rounds (see Gächter et al., 2008, for an example of comparing shorter-term versus longer-term dynamics).

Answer: We concur with R3 in raising this point because this is something we discussed, and an external view on this issue is very welcomed. As we explained in the manuscript, the fact that the games are iterated is essential for reducing potential free-riders that would pay a one-time cost and then defect during the single PGG iteration. Multiplying the costs through iterations, we reasoned, should repel free-riders. We selected three iterations as the least number of iterations that should have this effect while not prolonging the whole experimental procedure. Nevertheless, we agree that by adding iterations, the expected effect should be more substantial and, therefore, decided to raise the number of iterations to five. While not as high as the number of iterations in the paper suggested by R3 (Gächter et al. used 50 iterations), we believe that in contrast to the punishment scenario where each of the members may lose their full endowment, our participants lose only 15% of their endowment and the beneficial effects of this cost should appear earlier (compared to the cost of punishment).

Action: We increased the number of iterations to five and decreased the endowment for each iteration.

R3: Finally, one thing to consider is that, over time, real-life groups might need fewer and fewer costly signals to maintain coordination and cooperation. This would mean that, over time, group payoffs would increase in groups of signalers, as they would achieve cooperation with fewer and fewer costs. It seems that the design of the current study precludes studying this kind of process, because participants make a one-off decision to signal or not across all PGG rounds.

Answer: This is an exciting suggestion, and we believe that ritual costs may indeed start to decrease after some time due to sufficient interpersonal trust in a community. However, this decrease would need to assume stable selective pressures for cooperation, which is rather unlikely. Indeed, the frequency of costly rituals and similar demands indicating norm adherence has been shown to co-vary with warfare, which, presumably, pressures within-group cooperation and the related adherence to cooperative norms (Sosis, Kress, Boster, 2007).

It could be argued that some rituals (e.g., rites of passage) performed when accepting new members into a group are usually highly costly to probe the dedication of the 'unknown' members who, if successfully initiated, may later engage in regular less-costly rituals characteristic for the community. This example is indeed relevant for our current research, and allowing participants to decrease the ritual cost in subsequent rounds would provide a fruitful avenue for other possible follow-ups.

Action: If our manuscript proceeds to Stage-2 registered report, we would discuss this extension of our research in the discussion section. As we noted in the manuscript, we aim to establish a general experimental procedure that could be later easily replicated, extended, and tweaked to answer lingering questions from the theory. However, before conducting any extensions, we need to show the relevance of the most basic design first.

3. Statistical issues

R1: The analysis of (transformed) PGG contributions using beta regression should be adequate, although I do not fully understand the nature of the transformation the authors apply to the raw data. It would maybe be more straightforward to analyze those data with a binomial likelihood. Say you have a total of K CZK, and G of them are given to the public pot, we have $G \sim \text{binomial}(K, p)$. In theory, this is equivalent to $G/K \sim \text{beta}(a,b)$, but the authors would not have to do any transformations and the model also automatically takes care of differences in the number of trials, i.e. total CZK. This might be relevant here as we have different (remaining) total endowments across experimental conditions.

Answer: The transformation of the data necessary for the beta regression is only cosmetic since beta regression does not allow for "0" and "1" values, i.e., allow only for (0-1) interval. Hence, zeros and ones present in the data are transformed to values extremely close to zero or one such that this transformation would not impact the inference one can draw from the data. For a detailed explanation, please see Smithson and Verkuilen (2006). However, we now mention in the manuscript that in the case that more than 1/3 of participants' responses would be composed of 0 and 1, we would also consider fitting a zero-or-one inflated beta model (ZOIB) that does not require this transformation.

Regarding the suggestion to use binomial regression instead of beta regression, the problem is that binomial regression assumes independence between individual “trials”. In our case, this would mean that each 1 CZK would need to be assumed to be allocated independently of the other CZKs. Such assumed independence would provide over-confidently precise estimates of the between-condition differences and inflate its statistical significance (because the binomial model would assume 200 “trials” in the concealed and 180 trials in the revealed group in the pilot).

Action: We now clearly explain that the data transformation is just cosmetic and describes the conditions for using the ZOIB model.

R1: The authors propose a three-step procedure to analyze their data, where they first introduce their main predictor (experimental group) followed by a set of covariates and, lastly, hierarchical data structure. This procedure seems rather ad-hoc and not grounded in theoretical considerations. Omitting or including predictors without explicitly considering the hypothesized causal structure among them can lead to many inferential problems (see chapter 6 in McElreath, 2020, Statistical Rethinking). The authors should justify why they think certain variables need to be controlled for in a given set of analyses.

R2: Demographic variables such as education and economic status are included in the analyses. Do they authors have any expectations regarding these variables? Furthermore, could the authors justify the specific economic status question used?

Answer: Regarding the covariates that we aimed to assess (age, education, economic status), we decided to omit these variables from the planned data analyses. While our previous research showed that all three variables positively impact financial allocations to other players in economic games in small-scale societies, we agree that they are usually intercorrelated and may, therefore, introduce biases into our regression models. While pilot data showed a different dynamic between males and females in some of the tested relationships, we will aim to recruit a comparable ratio of men and women for all our conditions.

Action: We will collect data on age, sex, education, and economic status, but we will not use these variables directly in our regression models. We will retain these variables for downstream formal exploration with directed acyclic graphs (DAGs). We also removed these variables from regression models analyzing the pilot data.

R1: Also, hierarchical data structure -if present- should always be accounted for statistically, so there is no reason to omit random effects in the beginning. Instead, it is often helpful to first consider an “empty” model with hierarchical structure but no predictors to get an idea about how the variance is structured across different levels.

Answer: We are happy to follow R1’s guidance and keep the hierarchical structure of our data accounted for in all our models.

Action: We now let the intercepts vary by an individual in all of our models where it is justified.

R1: The authors plan to test H2 only in individuals choosing the concealed group. If we assume that group choice is both determined by cooperativeness and perceived costs, group choice would be a “collider” for the relationship between those variables. Using only data from the concealed group would effectively condition on the collider such that the relationship between cooperativeness and perceived costs might be biased compared to the whole population. The authors might want to consider the potential for a “collider bias” and/or run the analysis for all participants.

Answer: We agree that these three variables indeed are interrelated and may potentially bias our results. As a remedy, we reanalyzed our pilot data testing H2 on the full sample and found that the inference we can draw from these results did not need to change.

Action: We reanalyzed the models testing H2 on the full sample.

R1: There are a couple of problems in the description and mathematical notation for the analytical models in table 1. The outcome Variable Y_i in the binomial (logistic) models is whether a given individual i chose the revealed group or not (0/1). There is just 1 "trial" per participant, so $n=1$, and the goal is to model $p_i: Y_i \sim \text{binomial}(1, p_i)$. We connect the linear model to p_i using the logit link ($\text{logit}(p_i)=\dots$). Also, there is no additional error term (epsilon) in binomial models, as there is in Gaussian models. The same is true for Poisson and beta models, which also do not include an additional error. Lastly, what does it mean that there is a single parameter for the effect of sex, age, education, and economic status? Shouldn't there be a parameter for each effect?

Answer: The terms refer to a group of parameters, of course, not a single parameter. We realize we were not clear in specifying the parameters.

Action: We now clearly explain the notation of our regression models. We also fixed the notation for binomial models and removed the error terms.

4. Minor points

R1: I might have missed it, but is it possible to also upload the Z-tree code to OSF?

R2: Assessment would have been easier with access to the materials (original and/or translated) for the proposed studies. For example, I was unable to assess the comprehension questions. Could the authors provide materials?

Action: Our OSF repository now includes all the requested materials.

R1: Page 5, lines 40-42: In the revealed group, why do you expect higher individual (instead of group) payoffs? The theory predicts higher levels of cooperation that should, first and foremost, result in higher group payoffs. Individual payoffs in PGG might be maximized by free-riders who are paired with cooperators.

Answer: It is true that, in theory, higher average earnings in the revealed group might be caused by high-earning free-riders and low-earning cooperators. However, given that our design relies on iterated PGG, this would mean that cooperators would keep contributing to the common pool even after repeatedly being deceived by free-riders, which we deem as extremely unlikely. Furthermore, analyzing just group earnings would decrease the statistical power of our regression models. Finally, since the costly signaling theory is firmly rooted in the individual-level selection framework, analyzing group-level outcomes might suggest that we argue that costly signaling works through a group-selection mechanism.

R1: Pages 5-6: The authors switch between positive (e.g. “if H1-2 would be supported”) and negative (e.g. “but H3-4 would not be rejected”) formulations. In frequentist hypothesis testing, we can only really reject the H0, so I would suggest the authors might reformulate this section.

Answer: We agree and thank R1 for this suggestion.

Action: We reformulated the text to refer to hypotheses as supported or not supported.

R1: Page 8, line 16: Please provide some more details on how intentions are coded from participant rationales for choosing their group? What exactly is counted as “too costly”?

R2: The authors should provide more detail (possibly in the SM) of how they will analyze the written rationales indicated by participants for choosing their group. Currently, they report that they will look for mentions of high costs. Could they provide a series of key terms/phrases they will use during the coding of responses? Could they provide some examples of what this looked like in the pilot, using original and translated written rationales? Eventually, in the final analysis, other researchers should be able to independently reproduce the coding of rationales, so the authors should consider sharing original and/or translated rationales.

Action: We now provide all this requested information in the Supplementary Material.

R3: information on the trustworthiness scale is lacking as is a precise description of the cooperation measure. More specifically, it would be helpful if the authors include the items of the trustworthiness scale they used in the Supplemental Materials.

Answer: We decided against using the trustworthiness scale, and hence providing additional information seems unnecessary.

R2: Table 1, hypotheses 3 and 4 require revision. Currently, H3 and H4 refer to endowment and earnings in the “second PGG scenario” --- should H3 and H4 relate to either one or all of the three rounds in the PGG? If the authors plan to analyze the proportion contributed per round, should a random effect for the intercepts of each participant be included?

Action: This was an old format of the table referring to pilot hypotheses. We fixed this description.

R2: It is unclear how the authors will investigate signal cost and between-group conflict (Table 1, interpretations) --- could the authors elaborate?

Action: We now provide a specific design suggestion on how we would tackle this extension of our design.

R2: Make clear that the sacrificed 10% is discarded and not be available during the PGG.

Action: We now highlight this feature of our design at several places in the manuscript.

R2: Given that in the pilot a participant was excluded for indicating an unlikely age, it is perhaps worth stating what will be done with unlikely data (e.g. excluded or treated as missing data and using the proposed multiple imputation method). It is not necessary to indicate potential unlikely data for each variable, but signalling that this might be an issue might be worthwhile given it occurred with the pilot. Alternatively, the online questionnaire could be designed in such a way to avoid this issue.

Action: We restricted the online questionnaire to allow only values allowed by our design.

R2: Define "substantial effect" (Table 1, interpretations column).

Action: We re-worded this statement as “if the 95% CI for this estimate cross zero”

R3: With regards to the measure of cooperation, how exactly did participants make their decisions, e.g., could they choose to contribute any percentage of their endowment or did they have specific options such as contributing in increments of 10?

Action: We now explicitly state in the manuscript that the increments will be made by 1 CZK.

Appendix B

Dear Prof. Chambers,

We are glad that reviewers found our revisions satisfactory, and below, we detail our final revisions to the manuscript. There is one important change that we would like to announce before launching the data collection. Since in-person data collection that we have planned for this experiment could be launched at our university in October at the earliest (assuming there will not be another wave of Covid-19 in the fall), we decided to move the second stage of data collection (the Public Goods Game) from laboratory to an online environment. This change has no impact on our design or hypotheses, and we will use the same pool of participants as planned for the laboratory study. Furthermore, we will utilize online tools to simulate an actual presence in the laboratory (RA explaining the design through MS Teams where participants will be present as a group and will be allowed to ask questions). Since the rest of the experiment would originally entail individual participants sitting in their cubicles in the lab, we are confident that we can successfully simulate similar conditions using online tools (programmed in an online version of z-tree). Our laboratory has experience conducting such online studies; hence we opted for this variant to speed up the data collection. We have made necessary changes to the manuscript to include transferring the design from the laboratory to an online environment. Apologies for announcing this change so late, but we hoped that by the time our Stage-1 manuscript will be accepted, we would be able to run the study in the laboratory.

Best wishes,

Martin Lang.

R1: The authors have decided to include a behavioral measure of "cooperative phenotype" based on hypothetical PGG contributions instead of the self-report measure they used before. In general, I agree that behavioral measures are often more predictive of (real-world) behavior in other contexts, but I worry about the independence of this measure as an explanatory variable. It might not be surprising that individuals who indicate cooperative intentions in a hypothetical PGG will also behave more cooperatively in a real PGG. Furthermore, participants might even remember the answer given in the hypothetical scenario and might play a similar strategy in the actual game independently of their actual latent cooperative dispositions. The authors might want to comment on this potential issue.

Answer: First, we would like to note that this is indeed a behavioral measure because participants will be paid based on their decision (they just do not know which decision exactly). So, this is not only a "hypothetical" scenario. Second, the Fischbacher et al. procedure will be part of the online questionnaire that will precede the actual data collection in the laboratory by at least a week, so this delay makes it less likely that participants will remember their choices. Finally, participants will learn about other participants' choices only at the end of the laboratory session (second part of the experiment). Thus, their decision in the actual PGG will not be influenced by the Fischbacher et al. game results.

R3: The authors have improved their statistical notation, but there are still some issues. The link function in GLMs is not applied to the outcome but to a parameter in the likelihood. Focussing on the binomial model for H1: The outcome Y_i (0 or 1, not the probability as stated by the authors) is binomially distributed, so $Y_i \sim \text{binom}(1, p_i)$, " $n=1$ " because there is just one "trial" per participant. We then connect the participant-specific probability p_i to the linear model as $\text{logit}(p_i) = \dots$. In my view, including the model equations is also not absolutely necessary as long as the code is accessible and the models are correctly described, but if authors decide to include equations, they should make sure they are correct.

Action: We are grateful to R1 for correcting our equation, and we double-checked that the equations are correct now.

R2: 1) Could Figure S1 be moved to the manuscript? I find it a useful reference! 2) Could it be specified in H3, Table 1 that a "future" or "follow-up" study would investigate the effect of between-group competition? This would make clear that such a design would be warranted as part of a new study if the results of the current one are not confirmed (as the authors intend) and is not necessary as part of the current study. This future work could potentially consider other explanations, as group competition might be one of several plausible contexts under which the effects of costly signals may be observed.

Action: We made both suggested changes.

R3: In the revision, the authors suggest that it is not only extreme ritual behaviors that can signal a cooperative phenotype, but that more mundane behaviors, such as church attendance, can work in a similar way. This gives rise to the question of how costly a behavior needs to be in order to act as an honest signal of an underlying cooperative phenotype. I think that the authors' decision to manipulate the cost (high versus low) of signaling is a great addition to the design. But, it is possible with this design that the authors find support for their hypotheses, albeit not only for the high-cost condition. What would the authors conclude if, for example, participants with a cooperative phenotype are equally likely to signal in both the high- and low-cost conditions? Or if both low-cost and high-cost signaling are sufficient to boost contributions and stabilize cooperation? Would this go against the broader theory, or would it suggest that relatively low-cost signals might still allow assortment of cooperators (and maintenance of contributions at high levels)?

Answer: This is a great question. Observing that free-riders are deterred from entering the revealed group also in the low-cost condition would probably mean that other psychological factors (other than computing the cost/benefit ratio of signaling) are at play. This result would not go against the broader theory in general (after all, the low-cost condition is still costly) but would require further extensions to the theory to incorporate human-specific psychology.

Action: We have added this possibility to the paragraph in the introduction section, where we discuss possible results and their interpretations.

R3: What feedback will participants receive between rounds of the PGG? In the manuscript, the authors write that "Using their remaining endowment, participants in both groups will make first iteration PGG decisions and learn about their current earnings." Does this mean that participants will not receive information about other group members' contributions? If so, this might allow selfish individuals to 'fake' a cooperative phenotype by signaling, but keep making low contributions while remaining undetected (e.g., if they happen to be in a revealed group in which all other group members are high contributors).

Answer: Participants will receive information about other players' contributions (without knowing the actual identity of those players) after each round. So, they will have feedback on others' contributions that can inform their decisions in future PGG iterations. However, note that there is no punishment option, so once the free-rider would enter the revealed group, there is no way to pressure them by punishment to cooperate. This is another important point that we will raise in the discussion section of our Stage 2 manuscript.

Action: We revised the sentence to clarify that participants will see others players' contributions after each round.

Date 08/02/2022

Dear Prof. Chambers,

Following an in-principle acceptance of our Stage 1 registered report titled “Advertising cooperative phenotype through costly signals facilitates collective action” in *RSOS*, we are excited to submit the Stage-2 registered report that includes the results and discussion sections. After an in-principle acceptance, we closely followed our pre-registered methods and collected data (no data were collected before acceptance), and we used the pre-registered statistical models to analyse the data. We believe that the reported results offer several important insights into the role of costly signals in assorting co-operators and that the experimental framework we developed will of use to the scientific community, including our public materials, data, and statistical code. All of the above including the Stage-1 RR can be found on our OSF repository linked in the manuscript on p. 27.

For our submission of the Stage-2 RR, we provide both a ‘clean’ version of the manuscript as well as a version with tracked changes in the Introduction and Methods section. In general, these changes related to changing the tense from future to past and some minor clarifications or a different word choice.

We look forward to hearing from you and we are at your disposal for any further information.
On behalf of the authors,

Martin Lang

Masaryk University, Faculty of Arts, Department for the Study of Religions, LEVYNA

Arna Nováka 1/1, 602 00 Brno, Czech Republic

E: martinlang@mail.muni.cz, <http://levyna.cz>

Appendix D

Dear Prof. Chambers,

We are glad that reviewers found our STAGE 2 manuscript satisfactory, and below, we detail the requested revisions to the manuscript. There are some questions raised by R1 we were uncertain how to handle.

First, R1 suggests that we should update our models in Table 1 ("Relatedly, Table 1 does not seem to be updated, as there is still the formula for the regular beta regression."), but these models reflect our plan from the Stage-1 manuscript, and Table 1 explicitly mentions the contingency that we may use ZOIB instead of beta regression. We reasoned that Table 1 should reflect our thinking before seeing the data and therefore should not be updated.

Second, R1 also suggests moving the pilot results into supplements. We agree that this section is a bit lengthy and could be moved to supplements to streamline the final manuscript, but it was part of the Stage-1 manuscript. Our understanding is that we should not make this change.

Third, R1 also suggests a change to the title of our paper. The suggestion is helpful, but we again note that we should not make this change given the format of the registered report.

Thank you for your advice on these issues, we will take your lead on these particular edits.

Best wishes,

Martin Lang.

R1: First, I do not fully understand why the authors use statistical categorization of individuals into different cooperative strategies instead of a more direct behavioral measure such as the (conditional) PGG contributions as predictors to test H1 and H2. The mixture model results are interesting, but using them as predictors ignores a great deal of variation that is present in the raw data. It would at least be interesting to see the results if raw contributions are used and whether the results would stay the same. Relatedly, using the classifications as predictors in regression essentially treats uncertain estimates as observed data and omits the uncertainty about cluster assignment. Ideally, the full distribution of categorization estimates would be used as predictors in the same model (or is there no uncertainty in cluster assignment?!). However, I am aware that these types of analyses are maybe not straightforward and somewhat deviate from the state-1 RR, so feel free to ignore these suggestions. In any case, the authors might want to report some more details about the finite mixture model and discuss this issue of propagating uncertainty throughout the analysis.

Answer: R1 points to interesting extensions of our pre-registered analyses. Regarding the suggestion to use raw data on conditional cooperation rather than the classification, we note that raw contributions may be misleading because they do not necessarily reflect at what points the participant followed or did not follow the assumed mean contributions of other players (e.g., matched others' contributions in the low/middle/or upper mean contributions?). Moreover, the posterior probabilities of our classifications were all > 0.999 , suggesting well-separated categories. We now describe this posterior probability in the results section. Finally, the classification of participants into individual cooperative types indeed

assumes an underlying distribution for each type. It could be argued that some participants classified as selfish may be closer to tempted than other selfish participants. To account for this variation, we added an analysis into our Supplementary R code, where we use the raw contributions as predictors just for the selfish and cooperative types to account for this heterogeneity. The results match the results reported in the main text. We noted in the results section that this additional analysis is available in the R code.

R1: I also have some questions about the zero-or-one-inflated beta model. Mixture models are appropriate when there are different processes generating the data, i.e., zeros (or ones) can arise through multiple ways, not when zeros (or ones) are generated from the same process as other outcomes, even if there are many of them. Are the authors really assuming a mixture of different processes (which?) or are the results rather stemming from inter-individual differences. Do you assume, for example, that individuals who contributed either 0 or 1 did not understand the task in the same way?

Answer: As discussed in our reply to the previous comment (and to some extent in the next one), we work with a (simplistic) assumption of types of cooperative strategies that we classify using the finite mixture model. People falling into these cooperative types should behave differently in the public goods game – in an ideal case contributing zero to the common pool when classified as selfish and everything when classified as cooperators. Therefore, selecting these boundary contributions (0% or 100%) has a symbolic value that amplifies the assumed cooperative types: the difference between giving 0% and 10% would be perceived as more prominent than the difference between 5% and 15% (despite being the same difference). Modeling these sub-processes using a mixture model adds an important nuance to our main hypothesis regarding contributions to the common pool (H3). This point is further supported by the large number of zero and one contributions (larger than we expected), highlighting that these choices were salient for participants and corresponded to a distinct decision-making process.

Having said that, we also provide a visual illustration of the estimated portion of the remaining endowment allocated to the common pool in Figure 4C. As noted in the figure caption, this estimate is produced by the standard beta model that includes zeros and ones as originally planned and, therefore, gives readers a chance to compare the results from the standard beta models with the zero-or-one-inflated beta model that we report in the results section. We added a note into the figure caption that precise estimates from the standard beta model can be found in our Supplementary R code.

R1: As mentioned in previous reviews, I still find the term "the cooperative phenotype" rather problematic. Human cooperation is highly flexible and context-dependent and even though stable inter-individual differences surely exist, the trait reflected in those game choices hardly qualifies as THE cooperative phenotype, but rather corresponds to the tendency to show more cooperative behavior in the context of economic games played online with anonymous others. Additionally, the term phenotype suggests a single trait or unitary construct, whereas cooperative behavior is surely generated by a large suit of different processes. I would suggest the authors to use more careful wording that more closely corresponds to the actual behavioral measure or at least better qualify the term and/or use quotation marks.

Answer: R1 makes several points that we are sympathetic to, but we would like to defend our use of the concept of cooperative phenotype. First, we note that the use of "cooperative phenotype" in our manuscript reflects the broader use of this concept in the literature (Peysakhovich et al., 2014; Reigstad et al., 2017) as a general personality trait that is relatively stable. Second, using the concept of phenotype fits into the main aim of the current manuscript – testing the costly signaling theory, which predicts that phenotypic quality affects the probability of sending a costly signal.

However, we agree that "cooperative phenotype" has been operationalized in previous research solely through economic games, thereby decreasing its broader applicability. We already reflected this limitation in our last revision by talking about "cooperative strategy" rather than cooperative phenotype when referring to our measurements and directly discussed this issue in the limitations paragraph in the discussion section (as noted by R2). And, while it was written in the spirit of academic phrasing, it appears to have been misleading. To further clarify that the term is not used literally in our research, we added quotation marks to its first use in the introduction and discussion sections. To avoid any other confusion, we have also mildly softened this language (e.g., from "signalling the cooperative phenotype" to "signalling a cooperative phenotype") throughout.

Peysakhovich, A., Nowak, M. A., & Rand, D. G. (2014). Humans display a "cooperative phenotype" that is domain general and temporally stable. *Nature Communications*, 5, 1–8.
<https://doi.org/10.1038/ncomms5939>

Reigstad, A. G., Strømmland, E. A., & Tinghög, G. (2017). Extending the cooperative phenotype: assessing the stability of cooperation across countries. *Frontiers in Psychology*, 8(November), 1990.
<https://doi.org/10.3389/fpsyg.2017.01990>

R2: However, could the authors clarify the statement relating to the reduction in statistical power between the first (n = 372) and second (n = 284) phase of the study? The reduction is pointed out in the results (p22 l15), with reference to the a priori power analysis summarised in Figure S1. The authors state that 284 participants in total (i.e. 142 participants per condition) should be sufficient to detect the expected effects with 80% power. Is this assuming a moderate effect size? Figure S1 suggests 142 participants would achieve >80% power for H3 and ~80% for H4 (assuming a moderate effect size). But the actual sample size per condition is 156 (high cost) and 128 (low cost) participants -- does the lower bound of 128 participants not mean >80% power for H3 but ~75% for H4 (assuming a moderate effect size)? The reduction in sample size is not an issue: it is due to drop-outs, which are unavoidable given the approximately one week interval between phases, and the authors tried to mitigate by recruiting 138 more participants than required. However, the statement about power in the results should be made clearer and any adjustment made to the statement if necessary. Furthermore, descriptive statistics of the sample in the second stage could also be provided, to get a sense of how similar the samples are across phases. The descriptives should include SD_age and/or age range.

Answer: These are good points, and we made the requested changes.

R2: I was able to run and reproduce all H1--H4 analyses included in "Advertising_cooperative_V1.0.Rmd". I was unable to assess the codification of open responses about the reasons for selecting either the revealed or the concealed group, as only the original (in Czech) open responses are provided. If the authors deemed it appropriate, English translations could be provided in "Free-response_ratings.csv" to allow others to reproduce the codification. I leave it to the authors to decide whether translation would be best done either by them, through manual input in a web service, or using an R package. In line with best practice, a data dictionary should be included, describing e.g. columns B0--B20 in "Advertising_cooperative_V1.0.csv", levels in column SEX. Without a data dictionary it is not immediately clear what the variables or levels correspond to!

Answer: We added the data dictionary to our OSF repository and noted that we would be happy to provide English translations of open responses upon request.

R2: No unregistered exploratory statistical analyses are reported. While the hypothesis table explicitly stated agnosticism about comparisons of selfish individuals with tempted individuals, could the authors include such a comparison in the SI? This could guide future studies that might consider analysing this strategy --- it appears tempted individuals are e.g. less (more) similar to selfish (cooperative) individuals in mentioning a high cost.

Answer: We are reluctant to conduct these additional analyses because there is no strong theoretical reason to expect differences. The readers may gauge the raw differences from the tables in the main text. If desired, assessing whether these differences are statistically meaningful could be easily done using our publicly available R code and data set.

R3: When testing hypothesis 1, the authors do not find an interaction effect of low-cost × selfish strategy (the CI includes 0). Strictly speaking, this result does not provide support for hypothesis 1— i.e., the effect of cooperative phenotype on selecting the revealed group does not depend on whether the cost is high or low. In my opinion, this should be more explicitly noted. Nevertheless, the authors go on to interpret the simple effect of strategy in the high-cost condition, reporting that participants with a selfish strategy were less likely to chose the revealed group than the concealed group in the high-cost condition. Then, for the sake of completeness, I would also like to see the simple effect of selfish strategy in the low-cost condition.

- When testing hypothesis 2, the authors find support for the hypothesized interaction between low-cost × selfish on mentioning the costs of revealing. As before, they provide information on the simple effect of strategy in the high-cost condition. It would be helpful to also know what the simple effect of strategy was in the low-cost condition.

- When testing hypothesis 3, in two out of the three reported analyses, the interaction between low cost × concealed predicting contributions is not statistically significant (i.e., in the models looking at percent of contributions and the probability of sending everything). Again, my impression is that these results do not provide support for hypothesis 3 (as it was pre-registered)—i.e., the effects of being in the revealed group on contributions do not depend on whether the cost is high or low. This

could be more explicitly noted. Finally, the authors provide results for the effects of being in the revealed group on contributions in the high-cost condition, but not in the low-cost condition. For the sake of completeness, and to better understand the pattern of results in the two high- and low- cost conditions, it would be helpful to see simple effects in each of these conditions. The same applies to hypothesis 4.

In sum, I would like to see a more explicit mention of whether the pattern of results obtained in each analysis matches (or not) the hypotheses as pre-registered. It seems to me that, in many cases, the results do not provide support for the hypothesized interaction effects. Instead, there is support for simple effects consistent with the authors' ideas in the high-cost condition. Presenting the simple effects also in the low-cost condition would give readers a clearer idea about the full pattern of findings.

Answer: We appreciate R3's recommendation to be more precise in drawing inferences from our results, and we adapted the results and discussion sections accordingly. We now describe support for H1 as preliminary and in need of more testing and provide a possible explanation for the somewhat wider 95% CIs. We also edited inferences in the discussion section that were not based on the reported statistical tests. However, we disagree that we should proclaim our hypotheses as unsupported based on the level of p-value = 0.055 (H3). In our view, adhering to such artificial cutoff criteria (where supporting/rejecting a hypothesis may hinge on one data point) is detrimental. Finally, we do report the estimated differences between the revealed and concealed groups in the low cost condition, but we do not test whether these differences are statistically significant. We did not plan to assess these differences, and such an assessment would require post-hoc corrections for multiple testing, which would again deviate from our original plan. We leave it to interested readers to conduct these potential explorations using our open R code and data and use those explorations to inform their follow-up research.

Finally, we dealt with the various stylistic suggestions and minor suggestions that all reviewers had. All changes to the manuscript were tracked using the Microsoft Word track changes function.